# Metastability as a candidate neuromechanistic biomarker of schizophrenia pathology

**Fran Hancock**[1]*, **Fernando E. Rosas**[2,3,4,5], **Robert A. McCutcheon**[6,7], **Joana Cabral**[7,8], **Ottavia Dipasquale**[1], **Federico E. Turkheimer**[1]

1 Department of Neuroimaging, Institute of Psychiatry, Psychology and Neuroscience, King's College London, De Crespigny Park, London, United Kingdom, 2 Department of Informatics, University of Sussex, Brighton, United Kingdom, 3 Centre for Psychedelic Research, Department of Brain Science, Imperial College London, London, United Kingdom, 4 Centre for Complexity Science, Imperial College London, London, United Kingdom, 5 Centre for Eudaimonia and Human Flourishing, University of Oxford, Oxford, United Kingdom, 6 Department of Psychosis Studies, Institute of Psychiatry, Psychology & Neuroscience, King's College London, De Crespigny Park, London, United Kingdom, 7 Department of Psychiatry, University of Oxford, Oxford, United Kingdom, 8 Life and Health Sciences Research Institute School of Medicine, University of Minho, Braga, Portugal

* fran.hancock@kcl.ac.uk

**Editor:** Daqing Guo, Key Laboratory for NeuroInformation of Ministry of Education, School of Life Science and Technology, University of Electronic Science and Technology of China, CHINA

## Abstract

The disconnection hypothesis of schizophrenia proposes that symptoms of the disorder arise as a result of aberrant functional integration between segregated areas of the brain. The concept of metastability characterizes the coexistence of competing tendencies for functional integration and functional segregation in the brain, and is therefore well suited for the study of schizophrenia. In this study, we investigate metastability as a candidate neuro-mechanistic biomarker of schizophrenia pathology, including a demonstration of reliability and face validity. Group-level discrimination, individual-level classification, pathophysiological relevance, and explanatory power were assessed using two independent case-control studies of schizophrenia, the Human Connectome Project Early Psychosis (HCPEP) study (controls $n = 53$, non-affective psychosis $n = 82$) and the Cobre study (controls $n = 71$, cases $n = 59$). In this work we extend Leading Eigenvector Dynamic Analysis (LEiDA) to capture specific features of dynamic functional connectivity and then implement a novel approach to estimate metastability. We used non-parametric testing to evaluate group-level differences and a naïve Bayes classifier to discriminate cases from controls. Our results show that our new approach is capable of discriminating cases from controls with elevated effect sizes relative to published literature, reflected in an up to 76% area under the curve (AUC) in out-of-sample classification analyses. Additionally, our new metric showed explanatory power of between 81–92% for measures of integration and segregation. Furthermore, our analyses demonstrated that patients with early psychosis exhibit intermittent disconnectivity of sub-cortical regions with frontal cortex and cerebellar regions, introducing new insights about the mechanistic bases of these conditions. Overall, these findings demonstrate reliability and face validity of metastability as a candidate neuromechanistic biomarker of schizophrenia pathology.

**Data Availability Statement:** Human Connectome Project – Early psychosis Data and/or research tools used in the preparation of this manuscript were obtained from the National Institute of Mental Health (NIMH) Data Archive (NDA). NDA is a collaborative informatics system created by the National Institutes of Health to provide a national resource to support and accelerate research in mental health. Dataset identifier(s): NIMH Data Archive Digital Object Identifier (DOI) 10.15154/1528359. This manuscript reflects the views of the authors and may not reflect the opinions or views of the NIH or of the Submitters submitting original data to NDA. Cobre Cobre data is publicly available from the Collaborative Informatics and Neuroimaging Suite Data Exchange tool (COINS; http://coins.mrn.org/dx). Derived files MATLAB files for parcellated data for all subjects in both datasets is available at DOI: 10.5281/zenodo.7464484 Data and Code Data and code required to reproduce analyses, figures and tables is available at DOI: 10.5281/zenodo.7500799.

**Funding:** This study was funded by Wellcome Trust Career Development fellowship (https://wellcome.org/grant-funding/schemes/career-development-awards)(grant number 210920/Z/18/Z) awarded to RM. JC was supported by La Caixa Foundation (grant number project BRAINSTIM LCF/BQ/PR22/1192001) (https://fundacionlacaixa.org/en/) and by Fundação para a Ciência e a Tecnologia (https://3bs.uminho.pt/research-projects) (grant numbers UIDB/50026/2020, UIDP/50026/2020). Finally, FET and OD were supported by National Institute for Health Research (NIHR) Biomedical Research Centre (BRC) at South London and Maudsley NHS Foundation Trust and King's College London. The funders had no role in the study design, data collection and analysis, decision to publish, or preparation of the manuscript.

**Competing interests:** I have read the journal's policy and the authors of this manuscript have the following competing interests: RM has received honoraria for educational talks from Otsuka and Janssen.

## Introduction

Schizophrenia affects roughly 1% of the population, is associated with premature mortality and morbidity, and is accompanied by a large social and financial burden [1]. Originally described as the fragmentation of previously integrated mental experiences [2], the disorder is associated with positive symptoms such as delusions, hallucinations, and disordered thoughts, negative symptoms including amotivation and social withdrawal, and cognitive symptoms including deficits in executive function [3]. While schizophrenia can be a chronic disorder for a significant proportion of individuals [3], there is evidence that early diagnosis and treatment can lead to improved outcomes for patients [4].

The disconnection hypothesis of schizophrenia states that the disorder can be understood as a failure of functional integration in the brain. Functional integration is closely related with the functional connectivity, and with the influence of brain dynamics of one region on another [5, 6]. Failure of functional integration manifests as a disruption of the coordination required for the normal functioning of distributed brain regions [7]. For example, auditory verbal hallucinations have been associated with aberrant coupling in the speech processing system, speech production system, and the auditor monitoring system [8]. Additionally, amotivation has been linked with aberrant connectivity between the caudate nucleus and the cerebellum, leading to impaired goal achievement behaviour, and with prefrontal areas leading to poor goal-directed performance [9]. Moreover, disorganized symptoms have been predicted by aberrant connectivity between the cerebellum and the cingulo-opercular and salience networks [10]. Additionally, abnormal functioning of the basal ganglia in schizophrenia has previously been found with functional magnetic resonance imaging (fMRI) in schizophrenia [9, 11–13]. Indeed, ganglia hyperdopaminergia may be attributable to disconnectivity stemming from GABA parvalbumin interneuron disorder [14].

However, recent studies have highlighted differences in aberrant connectivity between early- and late-stage schizophrenia. Reduced cerebellum connectivity in early psychosis and increased connectivity in chronic schizophrenia were associated with both positive and negative symptom severity, suggesting a compensatory role for the cerebellum [15]. Disconnectivity between the somatosensory and visual networks was found to be pervasive in early psychosis, but not the disconnectivity between the default mode, cognitive control, and salience networks [16]. And finally, subcortical disconnectivity was found in early psychosis whilst both subcortical and cortico-subcortical disconnectivity was apparent in chronic schizophrenia. Importantly, the polarity of associations between disconnectivity and positive symptom severity were reversed for early and chronic groups, suggesting differences in neural correlates of psychotic symptoms at different stages of illness, and/or the potential effects of medication [17]. Hence, biomarkers of schizophrenia in early and established phases may differ, which may be informative of developing pathophysiology.

Disconnection in schizophrenia has been investigated with static functional connectivity (FC) [18–21]. However, static FC relies on statistical relationships between fMRI signals throughout the complete scan, which forces it to discard critical information about the brain's dynamics. In contrast, it is reasonable to believe that dynamic approaches–which consider the temporal dynamics of fMRI signals—may have the potential to discover more precise and informative biomarkers [16, 22–34]. Unfortunately, the literature provides no empirical studies investigating if approaches which rely on collective dynamical properties have better classification ability than those that rely on static FC properties, and whether dynamical approaches provide relevant insight for biological and cognitive interpretation.

To address this important issue, in this work we analyze the suitability of a specific marker of brain dynamics: *metastability*. Metastability is a concept originating from dynamical

systems theory which provides an explanation for the spontaneous and self-organized emergence and dissolution of spatiotemporal patterns of coordinated activity [35, 36]. In a neuroscientific context this reflects a tension established by the competition between trends for functional specialization and functional integration within and between brain regions [37]. Metastability is nowadays a ubiquitous concept across diverse models of brain functioning including coordination dynamics [38] and complex systems [39], while its metrics have found application in both empirical studies and computational modeling [40–49]. There are several reasons for choosing metastability as a marker of brain dynamics in schizophrenia. First, it reflects the competitive tension between integration and segregation, and is therefore relevant for studies based on the disconnection hypothesis. Second, commonly used dynamic metrics including group-level dwell/duration and occurrence/occupancy were found to differ significantly across multiple scanning sessions in healthy young adults, which could potentially blur state with trait variability [47]. Additionally, in that study, the reconfiguration process was found to be non-Gaussian, that is, there was memory in the network reconfiguration process. As such, it is not possible to use first-order Markov methods to calculate state transition probabilities. Moreover, the study concluded that global metastability was the only representative and stable metric of the 9 dynamic metrics investigated, highlighting its potential as a group-level biomarker of psychiatric disorders [47].

Building on this previous work, here we investigate how metastability would perform as a neuromechanistic biomarker of schizophrenia at the group- and individual-level; if this performance would carry over to face validation; what this putative biomarker would tell us about the pathophysiology of schizophrenia; and how well it could explain measures of integration and segregation.

We introduce a new measure for metastability as the mean variance of instantaneous phase-locking. Our rationale for this operationalization stems from the theory of Synergetics [50] and recent generalization of the Haken-Kelso-Bunz (HKB) model to multiple oscillators [51], which exhibits stable antiphase synchronization [52], and from the observation that differences in connectivity were not reflected in differences in the traditional measure for metastability within this study. We found that this novel proxy for metastability distinguished patients with established schizophrenia from healthy controls at the group-level with moderate effect size ($d$ = 0.77), delivered performance in the range of published individual-level classifiers for cross-validation, and out-of-sample testing, highlighted dysfunctional connectivity in basal ganglia in early schizophrenia, showed explanatory power of between 81–92% for measures of integration and segregation, and so demonstrated face validity of metastability as a candidate neuromechanistic biomarker schizophrenia pathology.

## Results

### Derivation of spatiotemporal patterns of phase-locking

We analyzed the resting-state fMRI activity from a total of 670 scanning sessions from the Human Connectome Project Early Psychosis (HCPEP) and Cobre datasets (see Materials and methods). In the HCPEP dataset healthy controls (CON, $n$ = 53) and subjects with non-affective psychosis (NAP, $n$ = 82) participated in 4 scanning sessions on 2 consecutive days. In the Cobre dataset CON ($n$ = 71) and subjects with schizophrenia (SCHZ, $n$ = 59) participated in 1 scanning session. Each dataset consisted of whole-brain fMRI signals averaged over $n$ = 116 cortical, subcortical, and cerebellar brain regions as defined in the AAL116 anatomical parcellation [53].

We used instantaneous phase-locking (*iPL*) to measure the interaction between fMRI signals related to different brain regions. The fMRI time-series of each subject was filtered within

the narrowband 0.01–0.08 Hz which did not violate the Bedrosian Theorem (see Materials and methods) [47]. The filtered signal was then transformed into amplitude and phase via the Hilbert transform, and the resulting phase time-series was analyzed via the Leading Eigenvector Dynamic Analysis (LEiDA) [47]. In order to identify recurrent spatiotemporal patterns of phase-locking–henceforth called 'LEiDA modes'–we performed k-means clustering on the phase-locked time-series of **each** of the datasets that were analyzed (HCPEP CONx4, HCPEP NAPx4, Cobre CONx1, Cobre SCHZx1, see Materials and methods). This is similar to a previous study [47], but different from other studies that used LEiDA where k-means clustering was either performed on concatenated datasets across groups [54–56] or where the centroids extracted from one group were used to seed the clustering of other groups [57–59]. The approach in this study considers each dataset as a unique observation of brain activity with associated variability in the spatiotemporal modes and avoids data leakage which occurs when dimensionality reduction is performed on the dataset as a whole [60]. We calculated the results for $k = 2$–10 clusters, and then chose $k = 5$ LEiDA modes—denoted as $\psi_1, \psi_2, \psi_3, \psi_4, \psi_5$— according to silhouette values [61] (see S1 Fig), which is consistent with previous studies [47, 54, 58]. Additionally, we calculated the instantaneous magnetization as the ratio of in-phase regions to antiphase regions, which indicates criticality [62]. Fig 1 shows the diversity of phase-locking behavior for two individual subjects from the HCPEP dataset.

We found that the 5 modes reflected connectivity within and across known resting-state networks, subcortical and cerebellar regions. Following Ref. [47], we visualized each mode in $10mm^3$ voxel space by averaging the eigenvector values over all time instances assigned to a particular mode. We visualized FC as connectograms by taking the FC matrices for each mode and retaining regions that were collectively in-phase but in antiphase with the global mode (see Fig 2).

Using the modes from RUN3 in CON as an illustrative example, we find that Mode $\psi_1$ represents a global mode where the fMRI signals in all regions are aligned in-phase without antiphase connectivity. Mode $\psi_2$ exhibits connectivity within Default Mode Network (DMN), Limbic network (LBC), and cerebellum (CB), and connectivity between DMN-LBC, DMN-subcortical (SC), DMN-CB, LBC-SC, LBC-CB. Mode $\psi_3$ shows connectivity within Somatomotor (SMT), Ventral Attention network (VAT), Frontal Parietal Area (FPA) and CB, and connectivity between SMT-FPA, SMT-CB, SMT-CB, VAT-FPA, VAT-SC, VAT-CB and FPA-CB. Mode $\psi_4$ exhibits connectivity within SC and CB, and connectivity between LCB-FPA, LBC-SC, LBC-CB, FPA-SC, FPA-CB, and SC-CB. Finally, Mode $\psi_5$ shows connectivity within Visual network (VIS), and between VIS-CB.

### Characteristics of spatiotemporal modes

Before assessing differences in the modes across the case-control groups, we first controlled if the modes observed in HCPEP were stable and representative across the four runs. We calculated run reliability within groups with interclass correlation ICC(1,1) [64] (See Materials and methods). The modes extracted for CON showed substantial to almost perfect reliability between runs with median ICC values $\psi_1$ (0.96), $\psi_2$ (0.98), $\psi_3$ (0.64), $\psi_4$ (0.89), and $\psi_5$ (0.77). The modes extracted for NAP also showed substantial to almost perfect reliability with median ICC values $\psi_1$ (0.97), $\psi_2$ (0.97), $\psi_3$ (0.96), $\psi_4$ (0.77), and $\psi_5$ (0.82) (see S2 Fig for all ICC matrices). We therefore confirmed that the modes tended to be invariant across multiple acquisitions in both case and control groups in HCPEP.

Concentrating first on HCPEP, we found that there was a strong contribution of basal ganglia regions to the leading eigenvector for Mode $\psi_4$ in CON. We therefore assessed if there were differences in basal ganglia connectivity, measured as contribution to Mode $\psi_4$, between

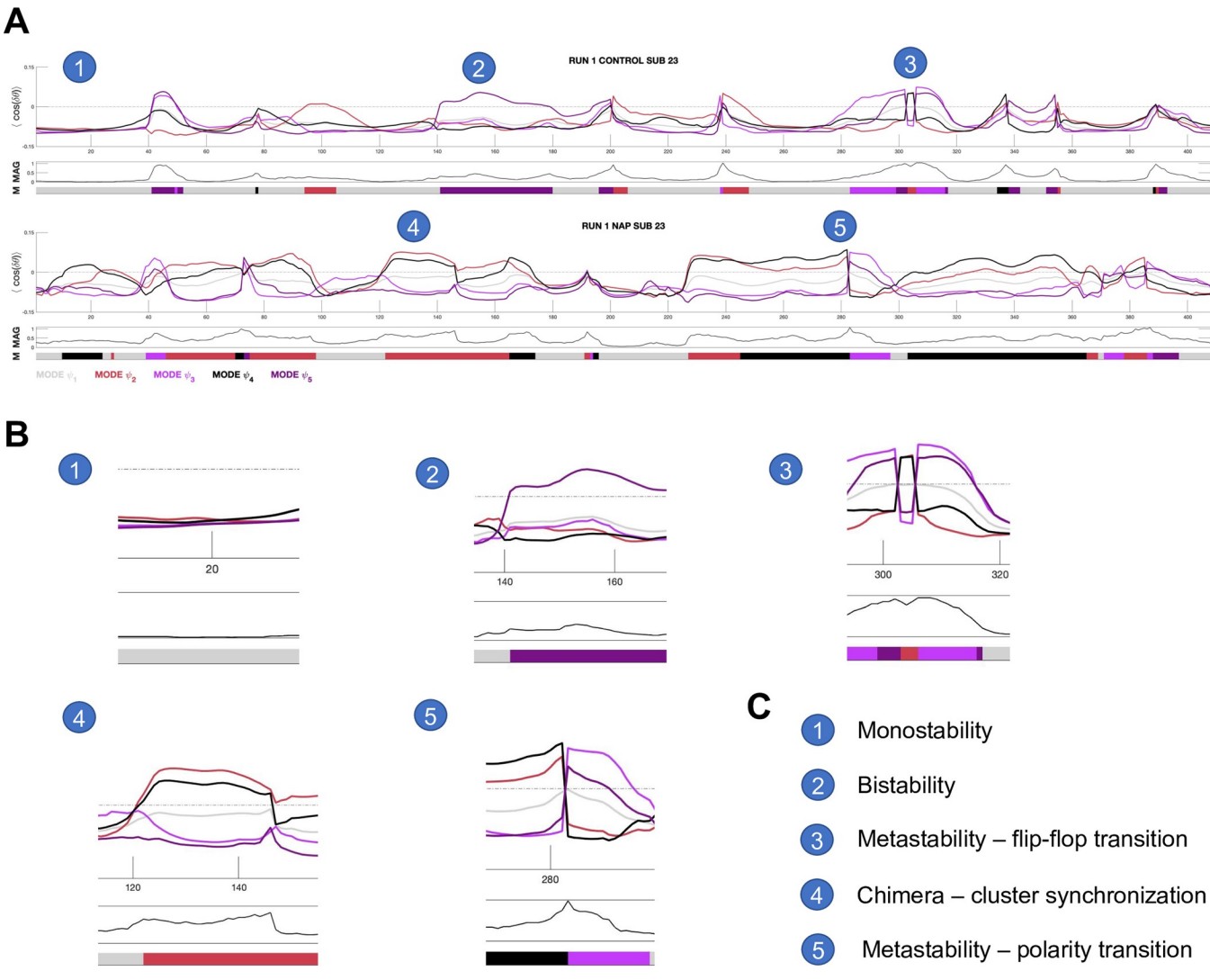

**Fig 1. Diversity of phase-locking behavior. A)** Time-series of mode eigenvectors from two subjects from the HCPEP dataset. Top panel shows phase-locking behavior. Middle panel shows instantaneous magnetization which is the ratio of in-phase to antiphase regions. Bottom panel shows the mode assigned to the timepoint from k-means clustering. Interesting behavior is indicated with numbered circles. **B)** Blow-outs for points 1 to 5. **C)** Legend for the numbered circles. MAG, magnetization; M, mode. Gray dotted line shows where phase-locking is equal to zero.

the groups. Regional contribution was calculated as the mean value of instantaneous phase-locking over time for the region of interest (ROI). We first investigated group (CON, NAP), run (RUN1, RUN2, RUN3, RUN4), and interactions between group and run on bilateral caudate, putamen, pallidum, and thalamus. Using a 2x4 non-parametric ANOVA with the Aligned Rank Transform (ART) [65, 66], we found significant interactions between group and run (Table 1).

We found significant main effects of run in both groups for multiple ROIs. The effects and the drivers of these effects are detailed in S1 Data. The largest main effects of run are shown in Table 1.

Furthermore, we found significant main effects of group in Caudate_L, Caudate_R, Putamen_L, Putamen_R, Pallidum_L, Thalamus_L, and Thalamus_R (Table 1). We retained only group differences that were greater than these run effects. We thus found significant group differences in RUN2 for Caudate_L ($p<0.001$, *effect size* = 0.435), Caudate_R ($p<0.001$, *effect size*

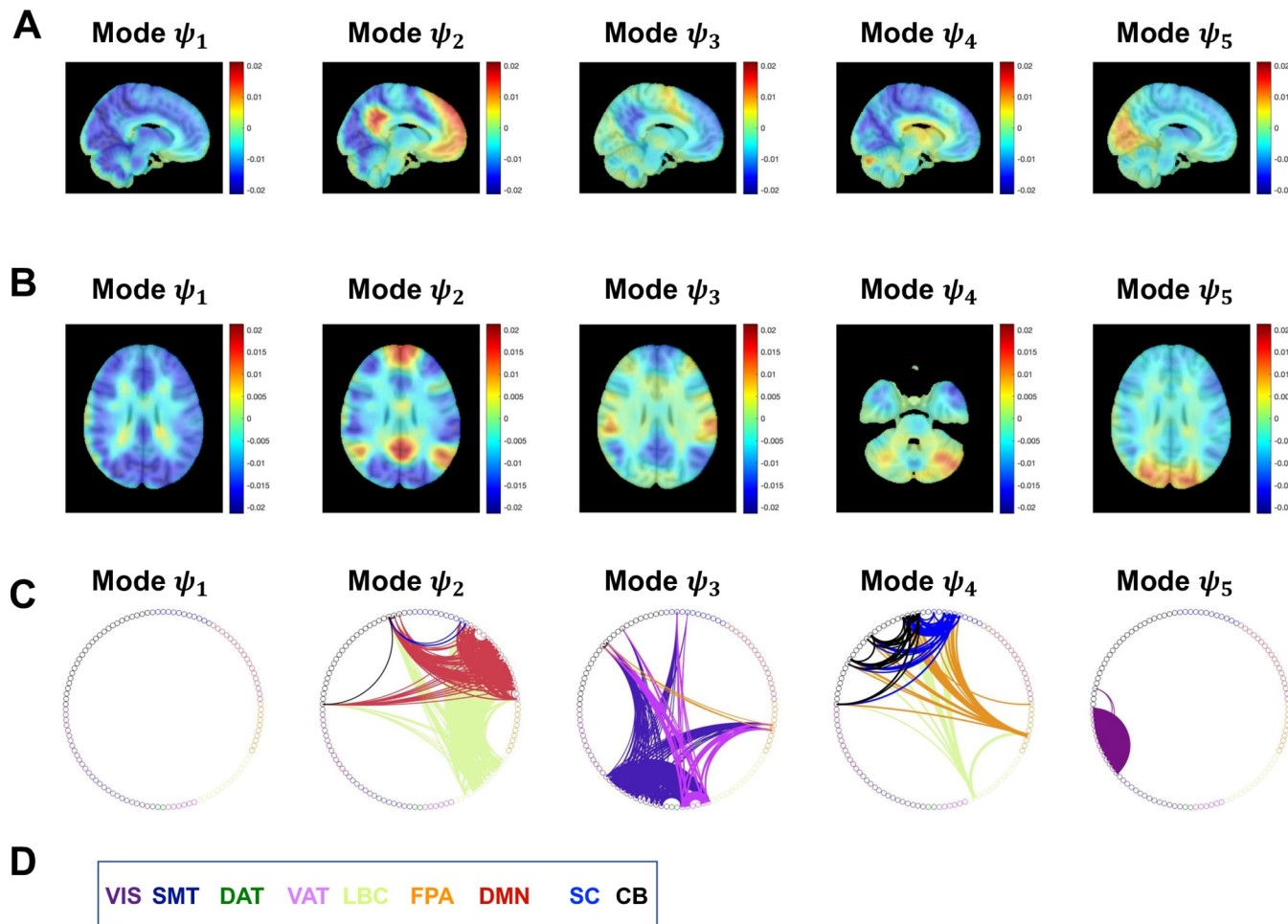

**Fig 2. Spatial patterns of recurrent phase-locked connectivity in RUN3 for controls. A**) Phase-locking patterns for the 5 modes in sagittal view. **B**) Phase-locking patterns for the 5 modes in axial view. **C**) Respective FC presented as connectograms color-coded as in Yeo [63] with the addition of dark blue for subcortical regions, and black for cerebellar regions. In Mode $\psi_1$ all regions are aligned in-phase and so there is no antiphase connectivity. FC computed as the outer product of the leading eigenvector for each mode. **D**) Color coded legend for the Yeo resting-state networks, subcortical and cerebellar regions. VIS, Visual; SMT, Somatomotor; DAT, Dorsal attention; VAT, Ventral attention; LBC, Limbic; FPA, Frontal parietal; DMN, Default mode network; SC, Subcortical; CB, Cerebellar.

= 0.523), Putamen_L ($p<0.001$, *effect size* = 0.351), Putamen_R ($p<0.001$, *effect size* = 0.357) and Thalamus_L ($p<0.001$, *effect size* = 0.526).

We therefore inferred that these group differences in basal ganglia contribution in RUN2 are not due to run effects, and indeed reflect group differences in regional contribution to Mode $\psi_4$. (See Fig 3 and S1 Data for complete results of the statistical testing).

## Global and local metastability–group-level neuromechanistic biomarkers of schizophrenia

To assess the performance of metastability at group-level, we computed and analyzed differences within and between groups based on the standard estimators for global and local metastability [67] (see S1 Text for the analysis, and S2 Data for complete statistical results). In contrast to previous studies of metastability in fMRI we choose not to use predefined templates [68], or intrinsic connectivity networks [41] to define our communities. The non-overlapping nature of these networks does not allow flexible allegiance of brain regions to different

**Table 1. Effects of group, run, and interactions between group and run, on contributions to mode $\psi_4$ connectivity in the bilateral caudate, putamen, pallidum, and thalamus.**

| Region of Interest | Main effect of group | | | Main effect of run (largest) | | | Interaction Group x Run | | |
|---|---|---|---|---|---|---|---|---|---|
| | Z | p | effect size | Z | p | effect size | F | | p |
| Caudate_L | 3296 | **<0.001** | 0.435 | 1179 | **<0.001** | 0.400 | F(3,339) = | 4.899 | **0.002** |
| Caudate_R | 3521 | **<0.001** | 0.523 | 656 | **<0.001** | 0.382 | F(3,339) = | 7.718 | **<0.001** |
| Putamen_L | 3079 | **<0.001** | 0.351 | 661 | **<0.001** | 0.335 | F(3,339) = | 7.923 | **<0.001** |
| Putamen_R | 3093 | **<0.001** | 0.357 | 668 | **<0.001** | 0.333 | F(3,339) = | 5.297 | **0.001** |
| Pallidum_L | 2804 | **0.005** | 0.245 | 1008 | **0.008** | 0.266 | F(3,339) = | 3.394 | **0.018** |
| Pallidum_R | 38175 | 0.054 | 0.083 | | | | F(3,339) = | 1.673 | 0.083 |
| Thalamus_L | 3530 | **<0.001** | 0.526 | 1132 | **0.001** | 0.360 | F(3,339) = | 8.105 | **<0.001** |
| Thalamus_R | 3038 | **<0.001** | 0.335 | 635 | **<0.001** | 0.349 | F(3,339) = | 3.032 | **0.029** |

Bold font indicates statistical significance following Bonferroni correction for multiple comparisons.

communities [69]. Rather, we defined the communities as the recurring spatiotemporal modes where individual brain areas may participate in more than one community. The number of brain regions in each community for each of the 10 datasets is shown in S1 Table.

We were somewhat surprised that metastability in Mode $\psi_4$ was not significantly different between groups in HCPEP given the differences found in basal ganglia connectivity. On reflection, we realized that the modes were extracted based on phase-locking, whilst metastability was computed on phase synchrony. In other words, the standard deviation of phase synchrony only captured the variability of the in-phase synchrony and ignored antiphase synchrony. Whilst this may seem to be a small methodological difference, it in fact highlights conceptual differences in the understanding of mechanisms of co-ordination across the brain [70] and the role of antiphase synchrony in large-scale cortical networks [71]. To rectify this methodological difference, we defined a new proxy for metastability as the mean variance of instantaneous phase-locking, VAR (See Materials and methods).

For the HCPEP dataset we first investigated group (CON, NAP), run (RUN1, RUN2, RUN3, RUN4), and interactions between group and run, on global VAR. Using a 2x4 non-parametric ANOVA with the Aligned Rank Transform (ART) [65, 66], we found a significant interaction between group and run (Table 2).

For the CON group, the effect of run was not significant. For the NAP group however, we found significant main effects of run, $\chi^2$ = 19.16, $p<0.001$, which were driven by significant differences in VAR between RUN1 and RUN3 ($p = 0.006$, *effect size* = 0.215), and between RUN1 and RUN4 ($p = 0.002$, *effect size* = 0.216).

Additionally, we found significant main effects of group which were driven by differences in VAR between CON and NAP in RUN1 ($p = 0.001$, *effect size* = 0.278) and RUN2 ($p = 0.002$, *effect size* = 0.263). As the effect size between groups in RUN1 and RUN2 were greater than the largest effect size between any pair of runs (Table 3), we inferred that metastability as measured with VAR differs between CON and NAP in RUN1 and RUN2. For the Cobre dataset, a permutation t-test for global VAR found a statistically significant difference between CON and NAP $t(126)$ = -4.17, $p<0.001$ for global VAR.

## Local metastability in the spatiotemporal modes

While global VAR reflects the average VAR across the modes, it is also of interest to assess the local VAR within the modes.

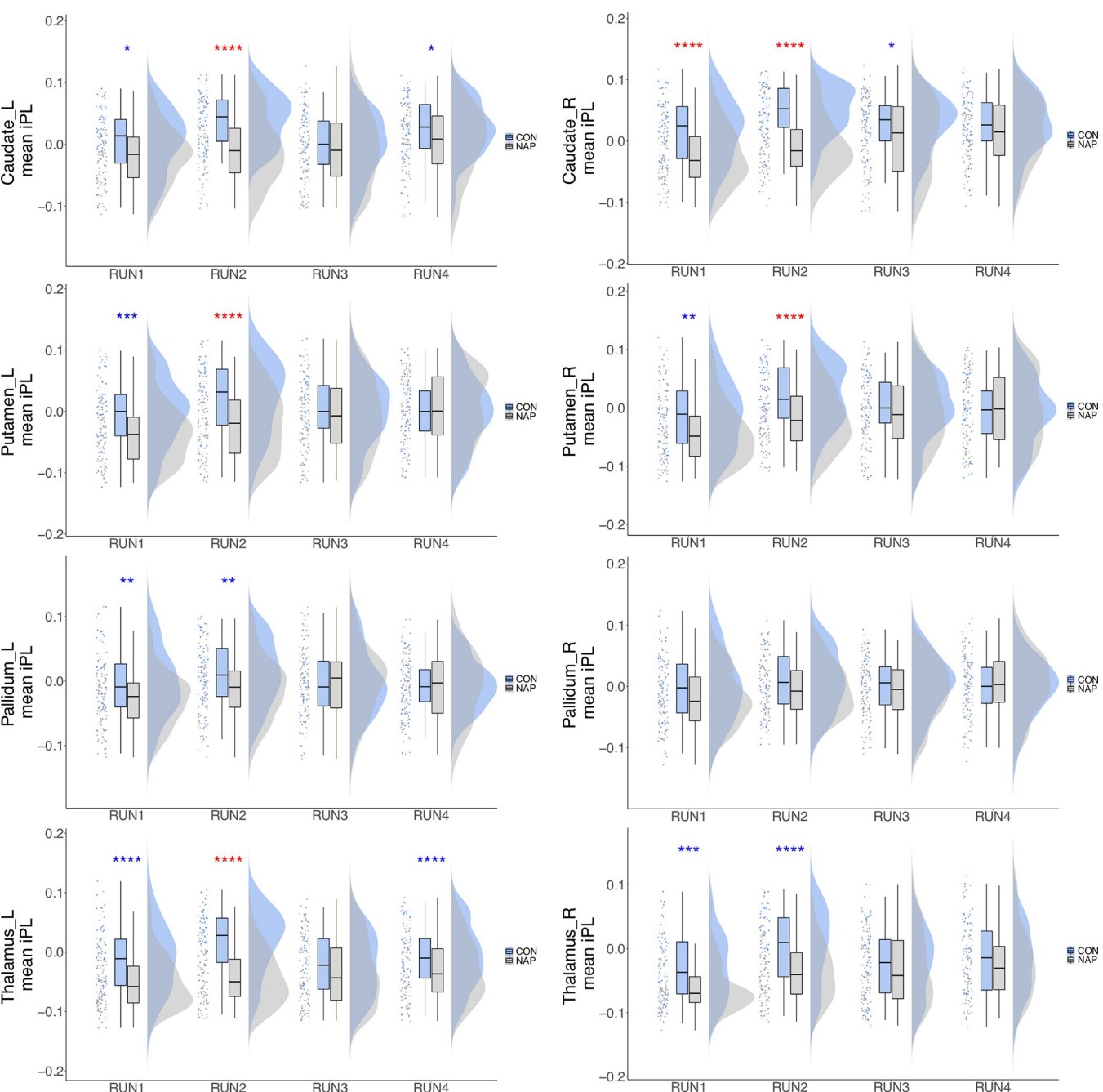

**Fig 3. Group differences in regional contribution to the leading eigenvector for Mode $\psi_4$.** Regional contribution was calculated as the mean value of instantaneous phase-locking over time for a particular anatomical region of interest. Raincloud plots show from left to right scatter plot for the raw data, boxplots showing the median, upper and lower quartiles, upper and lower extremes, and the distributions of the raw data. iPL, instantaneous phase-locking, $^*$ = 0.05, $^{**}$ = 0.01, $^{***}$ = 0.001, $^{****}$ <0.001. Red $^*$ effect size between groups greater than effect size between runs. Blue $^*$ effect size between groups less than largest effect size between runs.

For the HCPEP dataset we first investigated group (CON, NAP), run (RUN1, RUN2, RUN3, RUN4), and interactions between group and run, on global VAR for each mode $\psi_1$, $\psi_2$, $\psi_3$, $\psi_4$, $\psi_5$. Using a 2x4 non-parametric ANOVA with the Aligned Rank Transform (ART) [65, 66], we found significant interactions between group and run (Table 3).

**Table 2. Effects of group, run, and interactions between group and run, on global VAR.**

| | Main effect of group | | | Main effect of run (largest) | | | Interaction Group x Run | | |
|---|---|---|---|---|---|---|---|---|---|
| | Z | p | effect size | Z | p | effect size | F | | p |
| Global | 1455 | **0.001** | 0.278 | 2492 | **0.002** | 0.216 | F(3,339) = | 8.411 | **<0.001** |

Bold font indicates statistical significance.

In the CON group, we found significant main effects of run in $\psi_4$, $\chi^2 = 11.33$, $p = 0.010$. In the NAP group, we found significant main effects of run in $\psi_1$, $\chi^2 = 18.88$, $p<0.001$, in $\psi_2$, for $\chi^2 = 10.60$, $p = 0.014$, in $\psi_3$, $\chi^2 = 20.12$, $p<0.001$, and in $\psi_4$, $\chi^2 = 49.800$, $p<0.001$. The drivers for these effects and the associated effect sizes are detailed in S3 Data. The largest main effects of run are shown in Table 3.

Moreover, we found significant main effects of group in modes $\psi_1$, $\psi_2$, $\psi_3$, and $\psi_4$. The effect sizes of these differences were compared to the largest effect size between any pair of runs for that mode (Table 3). We thus found significant group differences for $\psi_1$ in RUN1 ($p = 0.007$, *effect size* = 0.234) and RUN2 ($p = 0.001$, *effect size* = 0.287), $\psi_2$ in RUN1 ($p = 0.003$, *effect size* = 0.258) and RUN2 ($p = 0.001$, *effect size* = 0.279), and in $\psi_4$ in RUN1 ($p<0.001$, *effect size* = 0.396, *moderate*) and RUN2 ($p = 0.001$, *effect size* = 0.399, *moderate*). We found a significant main effect of group for $\psi_5$, $p = 0.002$, *effect size* = 0.134.

We thus inferred that mode VAR differed between CON and NAP in $\psi_1$, $\psi_2$, $\psi_4$, and $\psi_5$ in RUN1 and RUN2, and in $\psi_5$ in all runs. For Cobre, we found statistically significant differences in mode VAR in all modes. (specifically, $\psi_1$ $t(125) = -3.423$, $p = 0.003$, $\psi_2$ $t(128) = -3.309$, $p = 0.007$, $\psi_3$ $t(124) = -3.584$, $p = 0.002$, $\psi_4$ $t(125) = -4.302$, $p<0.001$, and $\psi_5$ $t(128) = -4.745$, $p<0.001$). Complete statistical details for global and local VAR statistics can be found in S3 Data. Fig 4 shows the datasets with the most significant differences in mode VAR between groups.

## Relationship with neuropsychological processes

Based on the group-level results, and the results from our basal ganglia analysis, we now highlight the differences between CON and NAP in HCPEP for Mode $\psi_4$ in RUN2, and CON and SCHZ in Cobre for Mode $\psi_4$. To do so we compared the connectograms for each node and the associated behavioral topics from Neurosynth meta-analysis [72]. For the meta-analysis, we applied reverse inference to gain insights into potential behavior-relevant differences between cases and controls based on their spatiotemporal modes. Following the approach of [73], we used $t = 130$ terms, ranging from umbrella terms (attention and emotion) to specific cognitive processes (visual attention and episodic memory), behaviors (eating and sleep) and emotional

**Table 3. Effects of group, run, and interactions between group and run, on local VAR in modes $\psi_1$, $\psi_2$, $\psi_3$, $\psi_4$, $\psi_5$.**

| Mode of Interest | Main effect of group | | | Main effect of run (largest) | | | Interaction Group x Run | | |
|---|---|---|---|---|---|---|---|---|---|
| | Z | p | effect size | Z | p | effect size | F | | p |
| Mode 1 | 1434 | **0.001** | 0.287 | 2377 | **0.002** | 0.201 | F(3,339) = | 8.629 | **<0.001** |
| Mode 2 | 1453 | **0.001** | 0.279 | 2322 | **0.025** | 0.168 | F(3,339) = | 7.970 | **<0.001** |
| Mode 3 | 1548 | **0.005** | 0.242 | 2568 | **<0.001** | 0.278 | F(3,339) = | 6.395 | **<0.001** |
| Mode 4 | 1145 | **<0.001** | 0.399 | 2851 | **<0.001** | 0.351 | F(3,339) = | 20.250 | **<0.001** |
| Mode 5 | 29266 | **0.002** | 0.134 | | | | F(3,339) = | 2.005 | 0.113 |

Bold font indicates statistical significance.

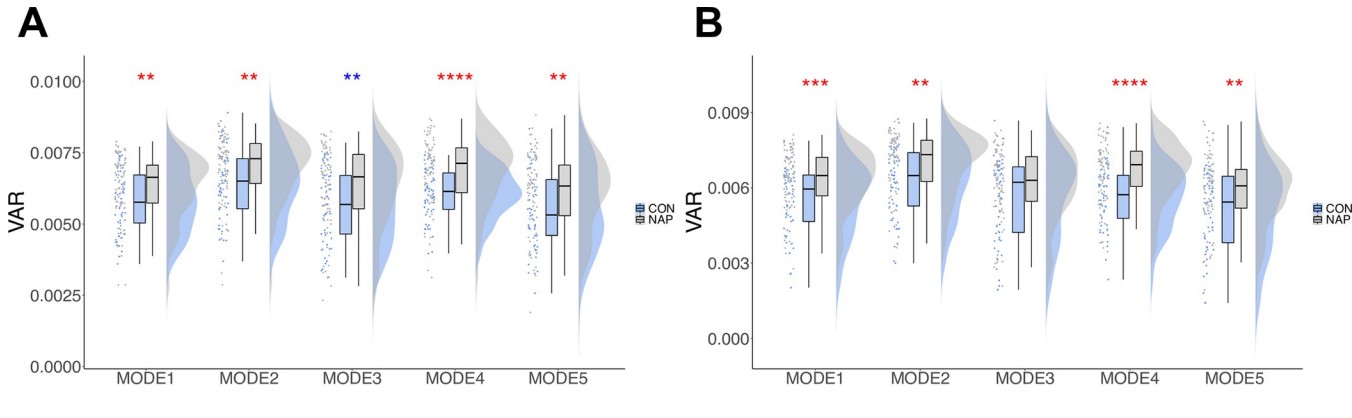

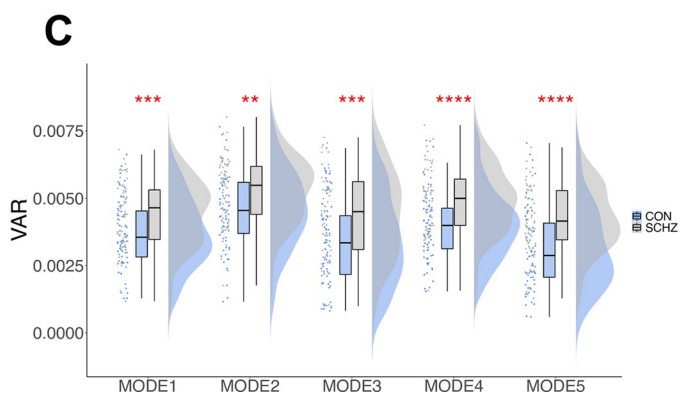

**Fig 4. Most significant group differences in local VAR in the modes for HCPEP and Cobre datasets.** Raincloud plots show from left to right the raw data, boxplots showing the median, upper and lower quartiles, upper and lower extremes, and the distributions of the raw data. **A)** HCPEP RUN1. **B)** HCPEP RUN2. **C)** Cobre dataset. * = 0.05, ** = 0.01, *** = 0.001, **** <0.001. Red * effect size between groups greater than effect size between runs. Blue * effect size between groups less than largest effect size between runs.

states (fear and anxiety). The coordinates reported by Neurosynth were parcellated into 116 cortical, subcortical, and cerebellar regions. The probabilistic measure reported by Neurosynth can be interpreted as a quantitative representation of how regional fluctuations in activity are related to psychological processes. We present the comparison in Fig 5.

We see from the meta-analytical terms in HCPEP that there is an absence of anticipation and reward anticipation in the NAP group compared with the CON group. In Cobre, fear, emotion, and anxiety are present in the SCHZ group but absent in the CON group.

### Global and local metastability–individual-level neuromechanistic biomarkers of schizophrenia

Our group-level results indicated that differences in VAR across groups were statistically significant in some modes, with effect sizes being small to moderate in HCPEP, and moderate to large in Cobre (S2 Table). We therefore decided to investigate the capability of these differences to classify subjects into their relevant groups. As VAR in Mode $\psi_4$ showed very large significant differences between groups in both HCPAP and Cobre, we decided to use this metric as an a-priori feature in a machine learning classifier.

Briefly, we chose a naïve Bayes classifier for its simplicity, but still compared its performance with other classifiers (as discussed below). We ran the classifier with repeated k-fold cross validation (*k* = 10, *repetitions* = 20) on balanced samples for training and cross-validation

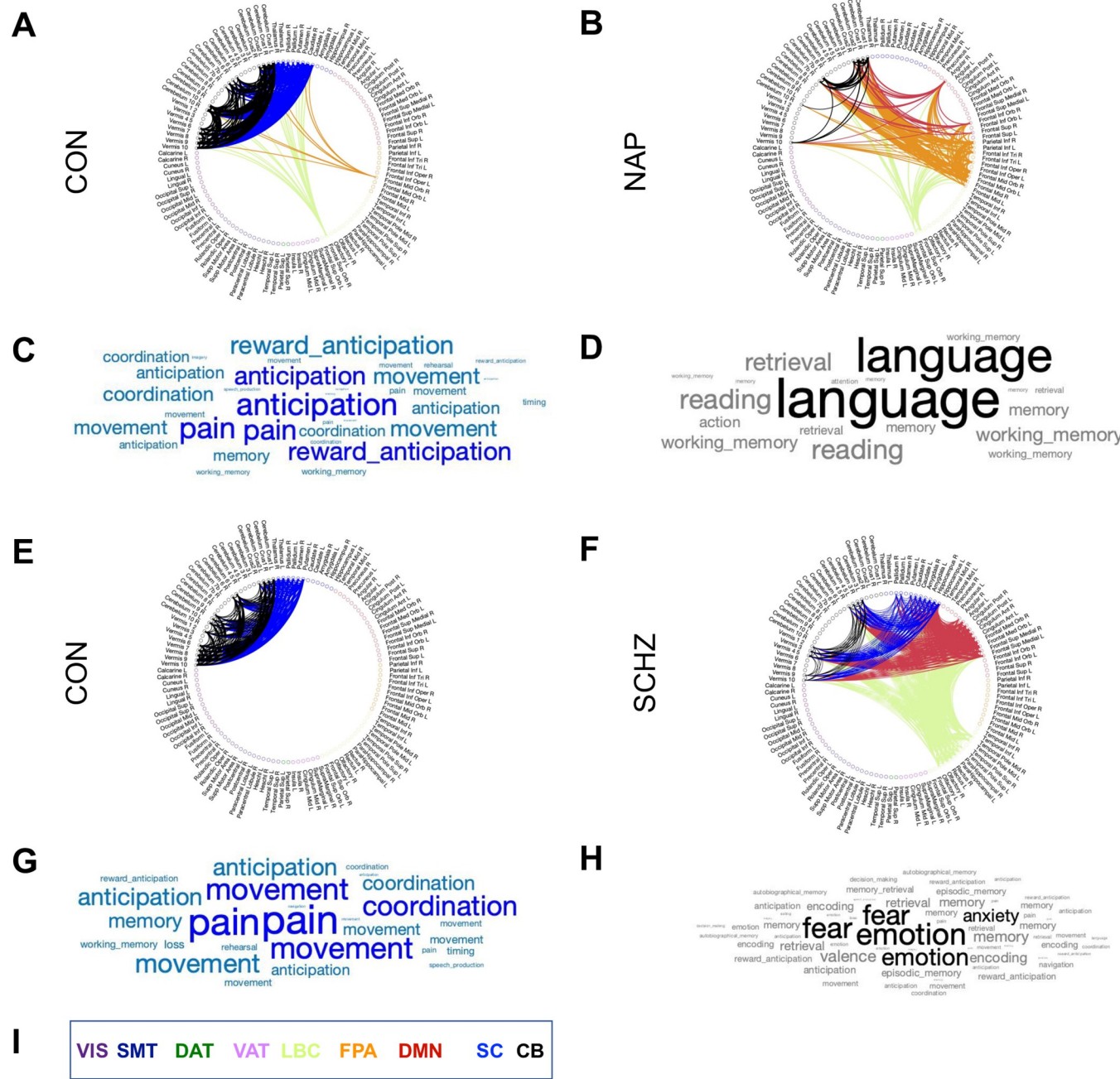

**Fig 5. Connectograms and word clouds for Mode $\psi_4$ in RUN2. A)** Group-level FC in Mode $\psi_4$ for HCPEP controls. **B)** Group-level FC in Mode $\psi_4$ for HCPEP Non-affective psychosis. **C)** The word cloud presents the top terms derived from Neurosynth using reverse inference for the regions in Mode $\psi_4$ for HCPEP controls. Word size represents the strength of the probabilistic association of the term to the regions. **D)** Top terms for Mode $\psi_4$ in HCPEP Non-affective psychosis. **E)** Group-wide FC in Mode $\psi_4$ for Cobre controls. **F)** Group-wide FC in Mode $\psi_4$ for Cobre Schizophrenia. **G)** Top terms for Mode $\psi_4$ in Cobre controls. **H)** Top terms for Mode $\psi_4$ in Cobre Schizophrenia. **I)** Color coded legend for the Yeo resting-state networks, subcortical and cerebellar regions. VIS, Visual; SMT, Somatomotor; DAT, Dorsal attention; VAT, Ventral attention; LBC, Limbic; FPA, Frontal parietal; DMN, Default mode network; SC, Subcortical; CB, Cerebellar.

in each dataset (HCPEP: 4 runs, Cobre: 1 run). Other than balancing the samples, we did not perform any preprocessing steps or tune hyperparameters. We then tested each classifier on an out-of-sample dataset, that is trained on HCPEP, tested on Cobre, or trained on Cobre, tested

**Table 4. Results of out of sample testing for each HCPEP run.**

| Train | Test | AUC | B. Accuracy | Sensitivity | Specificity | p-value |
|-------|------|-----|-------------|-------------|-------------|---------|
| RUN1 | Cobre | 0.37 | 0.38 | 0.17 | 0.60 | 0.004 |
| RUN2 | Cobre | 0.71 | 0.58 | 0.19 | 0.96 | 0.030 |
| RUN3 | Cobre | 0.59 | 0.55 | 0.66 | 0.43 | 0.058 |
| RUN4 | Cobre | 0.57 | 0.56 | 0.87 | 0.25 | 0.049 |
| **Train** | **Test** | **AUC** | **B. Accuracy** | **Sensitivity** | **Specificity** | **p-value** |
| Cobre | RUN1 | 0.74 | 0.50 | 0.96 | 0.04 | 0.080 |
| Cobre | RUN2 | 0.76 | 0.57 | 0.93 | 0.21 | 0.039 |
| Cobre | RUN3 | 0.52 | 0.51 | 0.96 | 0.06 | 0.080 |
| Cobre | RUN4 | 0.40 | 0.51 | 0.93 | 0.09 | 0.080 |
| **Train** | **X-Val** | **AUC** | **B. Accuracy** | **Sensitivity** | **Specificity** | **p-value** |
| RUN1 | RUN1 | 0.68 | 0.64 | 0.66 | 0.60 | 0.00 |
| RUN2 | RUN2 | 0.65 | 0.63 | 0.76 | 0.44 | 0.00 |
| RUN3 | RUN3 | 0.56 | 0.53 | 0.50 | 0.57 | 0.07 |
| RUN4 | RUN4 | 0.54 | 0.46 | 0.37 | 0.61 | 0.05 |
| Cobre | Cobre | 0.71 | 0.62 | 0.64 | 0.60 | 0.01 |

HCPEP RUN2 was chosen as the training sample for external validation in Cobre, and as the validation sample for classifier trained in Cobre.

on HCPEP (Table 4). HCPEP RUN2 performed best as measured by AUC when used as the training sample for out-of-sample testing in Cobre, and as the out-of-sample test for the classifier trained in Cobre. This implies that VAR in RUN2 captures best the feature that discriminates CON from NAP and SCHZ.

Although VAR in Mode $\psi_4$ was significantly different between groups in both RUN1 and RUN2, the superior performance of the classifier when trained in RUN2 may be explained by the effect sizes of basal ganglia decoupling in that run. Caudate_L (0.435), bilateral Putamen (0.351, 0.357), and Thalamus_L (0.526) showed medium effect sizes only in RUN2 (see S1 Data). The poor performance of the classifier when trained in RUN1, RUN3, and RUN4 can be explained by the non-pervasive decoupling of the basal ganglia.

We found that although the HCPEP classifier performed better than the Cobre classifier in cross-validation (HCPEP:$AUC = 0.73$, p<0.001, Cobre:$AUC = 0.71$, p = 0.007), the Cobre classifier performed better for out-of-sample testing (HCPEP:$AUC = 0.71$, p = 0.03, Cobre: $AUC = 0.76$, p = 0.039) as illustrated in Fig 6.

The superior performance of the classifier trained in Cobre can be explained by the different effect sizes of group-level differences in Mode $\psi_4$ between Cobre and HCPEP. In Cobre the effect size was -0.756 (see S2 Table), whilst the effect sizes in HCPEP varied across runs between 0.037 (RUN3) and 0.399 (RUN2) (see S3 Data).

In addition to a down-sampled naïve Bayes classifier, we also considered an up-sampled naïve Bayes, down-sampled Logistic regression, and a down-sampled Support Vector Machine models for VAR in mode $\psi_4$. Additionally, we used a down-sampled naïve Bayes classifier for META in mode $\psi_4$, for VAR in mode $\psi_4$ when calculated using NeuroMark [74] intrinsic connectivity networks, and for internal validation for HCPEP trained in RUN2 and tested in RUN1. The performance of these additional classifications is shown in Table 5.

As can be seen from Table 5, there was little difference in performance between down-sampling and up-sampling naïve Bayes, logistic regression, and Support Vector Machine classifiers. However, performance dropped significantly when the conventional metric for metastability, or a non-overlapping intrinsic connectivity network template, NeuroMark, were used in the classifier.

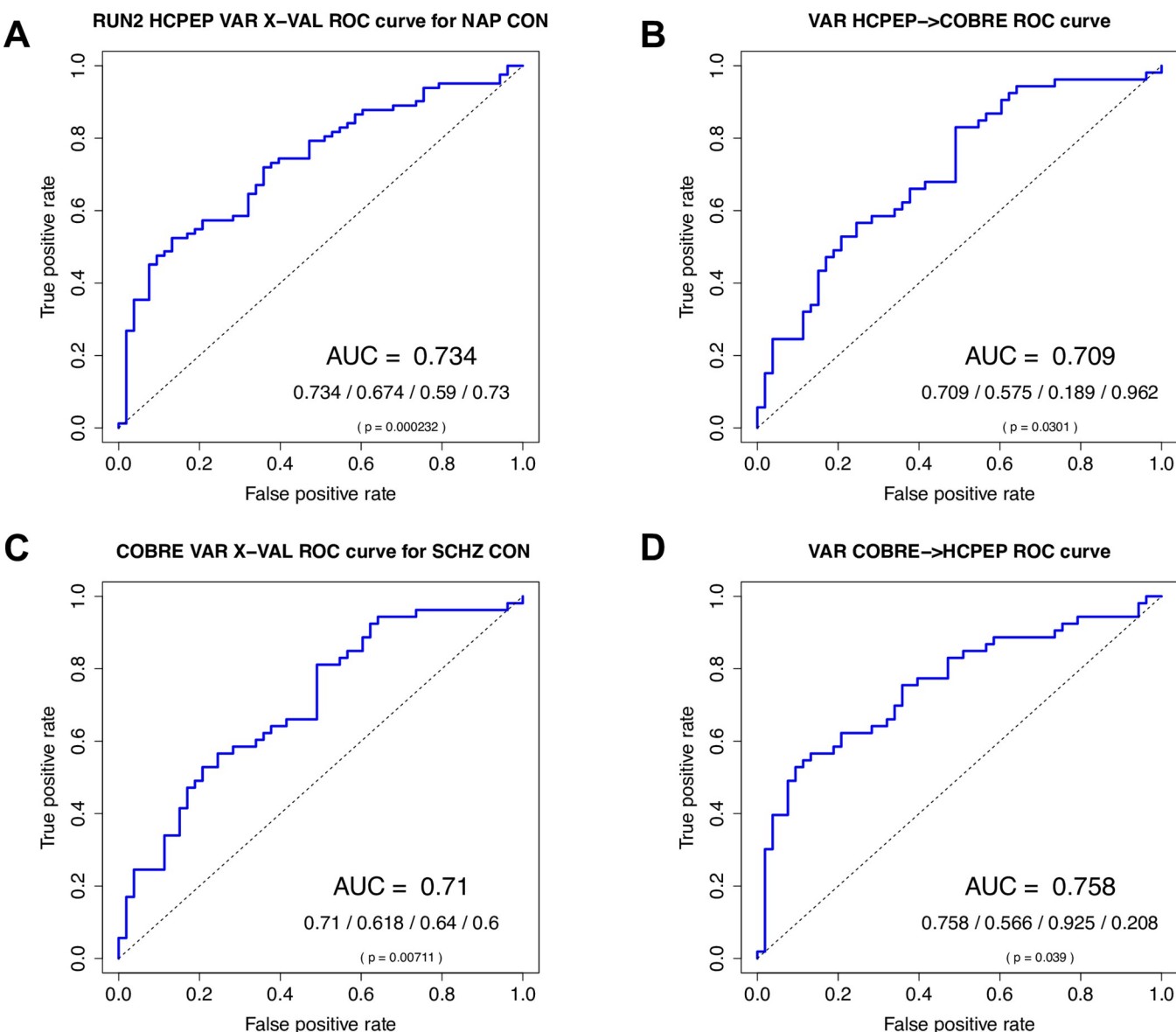

**Fig 6. naïve Bayes classifier results for discriminating cases from controls using a single a-priori feature VAR in Mode $\psi_4$. A)** Results for HCPEP model trained and cross-validated in RUN2. **B)** Results for HCPEP model trained and cross-validated in RUN2 and tested in Cobre. **C)** Results for Cobre model trained and cross-validated. **D)** Results for Cobre model trained and cross-validated in Cobre and tested in HCPEP RUN2. AUC/Balanced accuracy/Sensitivity/Specificity; p value calculated from the binomial distribution. AUC, area under receiver operating characteristic curve.

## Relationship between META, VAR, and measures of integration and segmentation

We have shown that VAR provides superior individual-level classification compared to the conventional metric for metastability, META. However, the classification ability sheds no light onto possible mechanistic explanations as to why this is the case. To address this question, we first need to understand that META and VAR are based on two different order parameters. In dynamical systems theory, an order parameter captures the collective behavior of an underlying high-dimensional non-linear system [39]. META is based on the Kuramoto order parameter [75] which is the mean phase in a system of weakly coupled oscillators. VAR on the other

**Table 5. Pserformance of additional classifiers in comparison to the classifier used in this study.**

| Metric | Classifier | Preprocessing | HCPEP X-VAL | | | | | OOS validation COBRE | | | | |
|--------|-----------|---------------|------|----------|-------------|-------------|---------|------|----------|-------------|-------------|---------|
| | | | AUC | Accuracy | Sensitivity | Specificity | p | AUC | Accuracy | Sensitivity | Specificity | p |
| *VAR* | *nBayes* | *downsampled* | *0.734* | *0.674* | *0.590* | *0.730* | *<0.001* | *0.709* | *0.575* | *0.189* | *0.962* | *0.030* |
| VAR | nBayes | upsampled | 0.744 | 0.672 | 0.591 | 0.726 | <0.001 | 0.709 | 0.585 | 0.208 | 0.962 | 0.022 |
| VAR | Logistic Reg | downsampled | 0.734 | 0.681 | 0.623 | 0.719 | <0.001 | 0.709 | 0.566 | 0.170 | 0.962 | 0.039 |
| VAR | SVM (linear) | downsampled | 0.734 | 0.675 | 0.606 | 0.721 | <0.001 | 0.709 | 0.575 | 0.189 | 0.962 | 0.030 |
| META | nBayes | downsampled | 0.518 | 0.490 | 0.649 | 0.388 | 0.078 | 0.530 | 0.528 | 0.491 | 0.566 | 0.074 |
| VAR:NM | nBayes | downsampled | 0.606 | 0.603 | 0.366 | 0.756 | 0.011 | 0.637 | 0.557 | 0.283 | 0.830 | 0.049 |
| | | | Cobre X-VAL | | | | | OOS validation HCPEP | | | | |
| | | | AUC | Accuracy | Sensitivity | Specificity | p | AUC | Accuracy | Sensitivity | Specificity | p |
| VAR | nBayes | downsampled | 0.710 | 0.618 | 0.640 | 0.600 | 0.007 | 0.758 | 0.566 | 0.925 | 0.208 | 0.039 |
| VAR | nBayes | upsampled | 0.710 | 0.618 | 0.638 | 0.601 | 0.007 | 0.758 | 0.566 | 0.925 | 0.208 | 0.039 |
| VAR | Logistic Reg | downsampled | 0.710 | 0.623 | 0.620 | 0.630 | 0.005 | 0.758 | 0.566 | 0.925 | 0.208 | 0.039 |
| VAR | SVM (linear) | downsampled | 0.710 | 0.625 | 0.631 | 0.662 | 0.005 | 0.758 | 0.575 | 0.943 | 0.208 | 0.030 |
| META | nBayes | downsampled | 0.650 | 0.630 | 0.593 | 0.668 | 0.003 | 0.526 | 0.538 | 0.396 | 0.679 | 0.067 |
| VAR:NM | nBayes | downsampled | 0.656 | 0.602 | 0.665 | 0.544 | 0.011 | 0.678 | 0.613 | 0.925 | 0.302 | 0.007 |
| | | | HCPEP X-VAL RUN2 | | | | | Internal validation HCPEP RUN1 | | | | |
| | | | AUC | Accuracy | Sensitivity | Specificity | p | AUC | Accuracy | Sensitivity | Specificity | p |
| VAR | nBayes | downsampled | 0.734 | 0.674 | 0.590 | 0.730 | <0.001 | 0.735 | 0.617 | 0.744 | 0.491 | 0.007 |

hand, is based on relative phase which has its origins in Superconducting Quantum Interference Device (SQUID) array experiments in the early 1990's [76]. To compare the relevance of both order parameters, we plot their time-series for one NAP subject, including measures of magnetization ratio, proxy for criticality [62] and chimerality or cluster synchronization [67] in Fig 7.

It can be seen from Fig 7 that taking in-phase and antiphase synchrony into account is more informative on the dynamics of brain activity than just in-phase synchrony, and the magnetization ratio is more relevant to mode switching than chimerality.

Metastable dynamics reflect the competitive tension between global integration and functional segregation [77]. Therefore, any metric or signature of metastability should be correlated with measures of integration and segregation. We calculated the level of global integration and functional segregation as in [78] and compared across groups. In HCPEP a permutation t-test for global integration (GINT) found a statistically significant difference between CON and NAP $t(100) = 3.70$, $p < 0.001$, and for functional segregation (FSEG) $t(109) = -3.14$, $p = 0.0023$. For Cobre, a statistically significant difference was also found for GINT $t(127) = 2.96$, $p = 0.0032$, and for FSEG $t(116) = -3.10$, $p = 0.0026$ (see Fig 8). Additionally, we calculated a Metastability index K, as segmentation/integration, in the spirit of [77]. We performed linear regression of the metrics for global integration, functional segregation, and Metastability Index on META and VAR in HCPEP RUN2 and Cobre. Complete statistical results may be found in S3 Table. For META in HCPEP the highest explanatory power, R2 adjusted, was 0.07 for NAP for global integration and in Cobre 0.55 for CON for global integration. In contrast, for VAR in HCPEP the highest explanatory power, R2 adjusted, was 0.91 for NAP for functional segregation and in Cobre 0.92 for SCHZ for functional segregation. The relationship between META and VAR with these 3 metrics are shown in Fig 8. Based on these results we can infer that VAR is a true measure of the competitive tension between global integration and functional segregation, and therefore a valid signature of metastability.

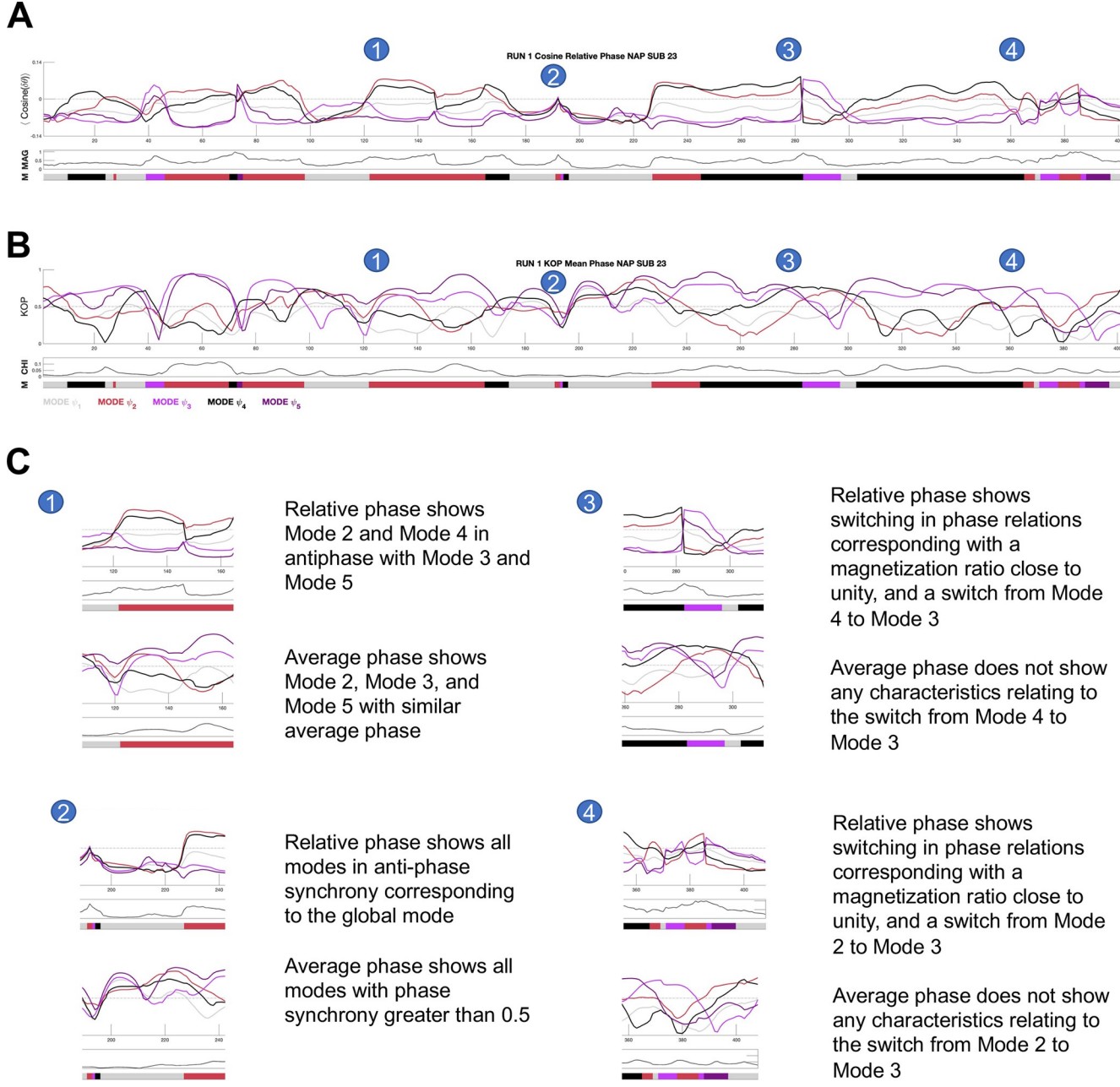

**Fig 7. Time-series for the relative phase order and the Kuramoto order parameter for one NAP subject. A)** Time-series for the cosine of relative phase or instantaneous phase-locking for a single subject. **B)** Time-series for the Kuramoto order parameter or average phase for a single subject. C) Blow-outs showing how relative phase is more informative than average phase for the dynamics of brain activity in one subject. MAG, magnetization ratio; KOP, Kuramoto order parameter, CHI, chimerality.

We have found that metastability as measured with VAR outperforms the conventional measure of META. We have shown that the order parameter of relative phase, as captured by instantaneous phase-locking which reflects in-phase and antiphase synchrony, is more informative than average phase for understanding the dynamics of brain activity. Finally, we have demonstrated that VAR explains between 81–92% of the variance in global integration, functional segregation, and the Metastability Index in the HCPEP and Cobre datasets. In summary,

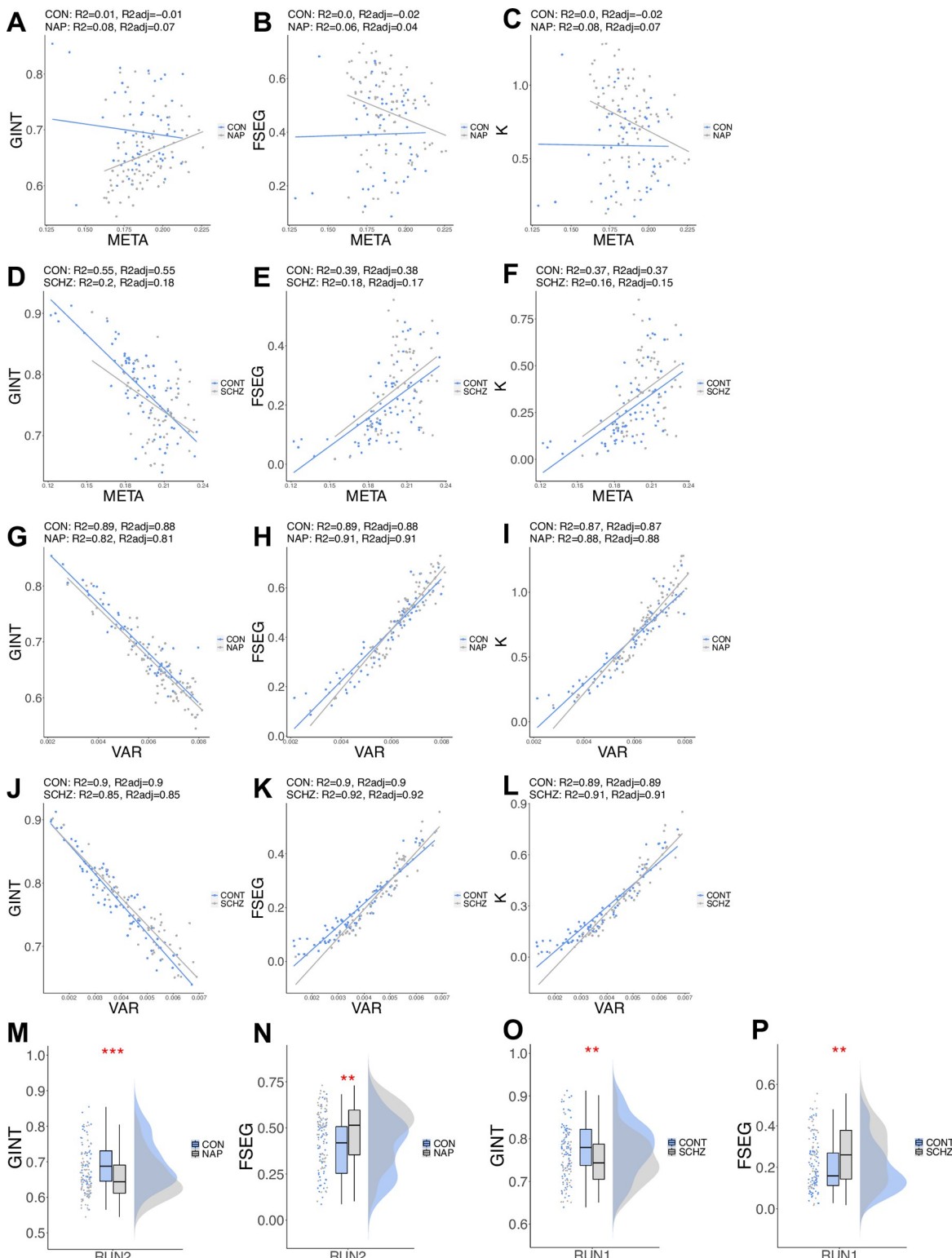

**Fig 8. Relationship between VAR and META with metrics of global integration, functional segregation, and a metastability index K.**
**A)** HCPEP global integration versus META. **B)** HCPEP functional segregation versus META. **C)** HCPEP Metastability Index versus META. **D)** Cobre global integration versus META. **E)** Cobre functional segregation versus META. **F)** Cobre Metastability Index versus META. **G)** HCPEP global integration versus VAR. **H)** HCPEP functional segregation versus VAR. **I)** HCPEP Metastability Index versus VAR. **J)** Cobre global integration versus VAR. **K)** Cobre functional segregation versus VAR. **L)** Cobre Metastability Index versus VAR. R2

and R2adj results from linear regression. **M)** HCPEP Group-level differences in global integration. **N)** HCPEP Group-level differences in functional segregation. **O)** Cobre Group-level differences in global integration. **P)** Cobre Group-level differences in functional segregation. GINT, global integration; FSEG, functional segregations; K, Metastability Index.

we have shown that antiphase synchrony is not only important for large-scale cortical networks [71], but also for characterizing metastability in healthy controls and groups with a diagnosis of schizophrenia.

## Discussion

In this study we set out to assess the face validity of metastability as a candidate neuromechanistic biomarker of schizophrenia. Our results provide preliminary evidence to support the premise that metastability measures dysfunctional connectivity in schizophrenia from 4 complementary perspectives.

First, we found statistically significant differences in group-level metastability between healthy controls and subjects with schizophrenia. Effect sizes were negligible to small ($d$ = -0.11 to $d$ = 0.36) for early disorder subjects (NAP group) and moderate to large ($d$ = -0.58 to $d$ = -0.82) for subjects with established schizophrenia (SCHZ group) (see S2 Table). Previous discrimination analysis on the same Cobre dataset using a distance measure between patterns of instantaneous phase synchrony reported a moderate effect size ($d$ = 0.67) [33], as did another study using the mean probability of dwell time in a global state (Hedge's $g$ = 0.73) [56]. In contrast, one study reported significantly lower effect sizes ($d$ = 0.06 to $d$ = 0.31) using measures of metastability in its original form, and using measures of between-network FC ($d$ = 0.04 to $d$ = 0.52) [68]. Although there are many studies that assess group-level differences in dFC, few report effect size. Therefore, limited to this small comparison, we consider that metastability, when calculated as the mean variance of instantaneous phase-locking, performs better than alternative group-level metrics reported in the literature.

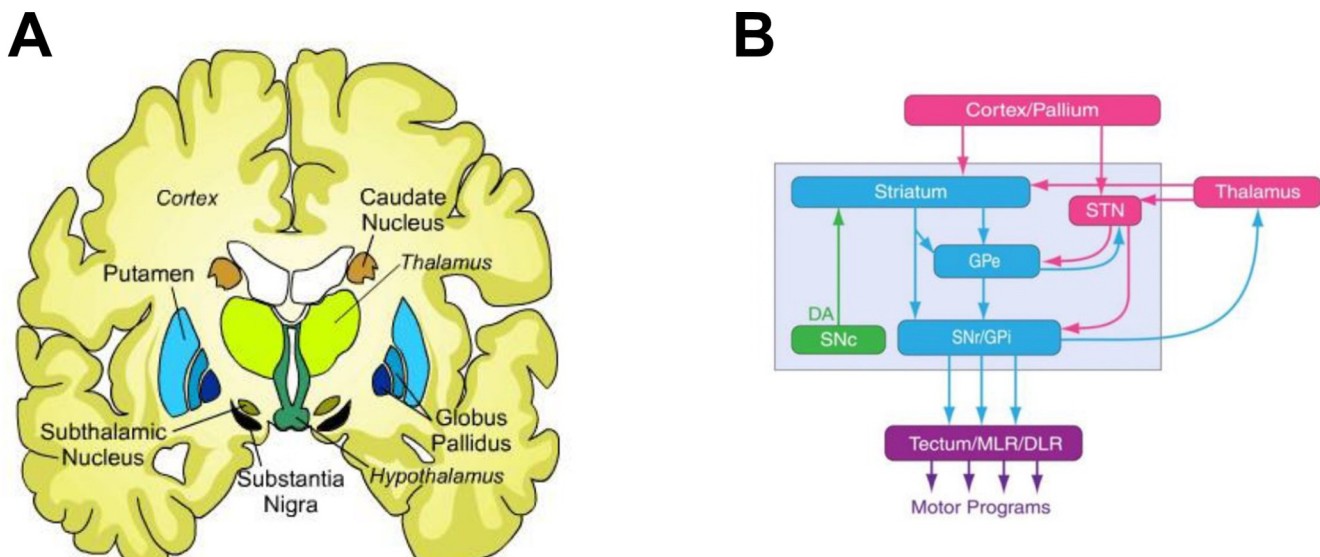

**Fig 9. Simple scheme of basal ganglia connectivity. A)** Location of the basal ganglia in an axial cartoon view of the brain. **B)** Basal ganglia connectivity. Arrows indicate direction of connectivity. Glutamatergic (Glu) structures are shown in rose, GABAergic nuclei are shown in cyan, and the dopaminergic (DA) nucleus is shown in green. STN, subthalamic nucleus; SNC, substantia nigra pars compacta; GPe, global pallidus external; GPi, global pallidus internal; SNr, Substantia nigra; MLR, midbrain locomotor region; diencephalon locomotor region.

**Table 6. Comparison of classifier performance using FC and dFC.**

| Cross validation | | | | | | | | | | |
|---|---|---|---|---|---|---|---|---|---|---|
| **Study** | | **Dataset** | **Controls** | **Cases** | **# features** | **AUC** | **B Accuracy** | **Sensitvity** | **Specificity** | **Comments** |
| Lei et al. (2020) | FC | COBRE | 72 | 68 | 4,095 | | 0.82 | 0.69 | 0.94 | |
| Morgan et al. (2021) | FC | COBRE | 73 | 60 | 42,778 | 0.75 | 0.70 | 0.62 | 0.77 | |
| Hancock et al. (2022) | dFC | COBRE | 71 | 59 | 1 | 0.71 | 0.62 | 0.64 | 0.60 | Case Positive |
| | | | | | | | | | | Downsampled |
| Morgan et al. (2021) | FC | Maastricht | 53 | 59 | 42,778 | 0.74 | 0.65 | 0.77 | 0.59 | |
| | | Dublin | 72 | 25 | 42,778 | 0.82 | 0.86 | 0.50 | 0.97 | |
| Rashid et al. (2016) | dFC | Hartford | 135 | 87 | 15 | | 0.84 | 0.83 | 0.92 | |
| Du et al. (2020) | dFC | BSNIP-1 | 238 | 113 | >1'000 | | 0.69 | 0.66 | 0.73 | |
| Hancock et al. (2022) | dFC | HCPEP | 53 | 82 | 1 | 0.73 | 0.67 | 0.59 | 0.73 | Case Positive |
| | | | | | | | | | | Downsampled |
| **External validation** | | | | | | | | | | |
| **Study** | | **Dataset** | **Controls** | **Cases** | **# features** | **AUC** | **B Accuracy** | **Sensitvity** | **Specificity** | |
| Morgan et al. (2021) | FC | Maastricht->Dublin | 53 | 59 | 42,778 | 0.77 | 0.56 | | | |
| | | Dublin->Maastricht | 72 | 25 | 42,778 | 0.76 | 0.69 | | | |
| Hancock et al. (2022) | dFC | COBRE->HCPEP | 53 | 53 | 1 | 0.76 | 0.57 | 0.93 | 0.21 | Case Positive |
| | | HCPEP->COBRE | 53 | 53 | 1 | 0.71 | 0.58 | 0.19 | 0.96 | Case Positive |
| | | | | | | | | | | Random Sampling |

Second, group-level differences in metastability (as measured with VAR) revealed group-level differences in dFC for both early and established schizophrenia. Specifically, intermittent functional disconnectivity was found for bilateral caudate, putamen left, and bilateral thalamus in early schizophrenia. The caudate and putamen are part of the dorsal striatum which is a key component in the basal ganglia. Fig 9 shows a very simple scheme of basal ganglia connectivity with the thalamus and cortex highlighting the substantia nigra pars compacta (SNc) which is the source of the neurotransmitter dopamine.

Elevated dopamine synthesis and storage have been implicated in the pathophysiology of schizophrenia [79]. Hyperactivity of the substantia nigra was found to be associated with decreased prefrontal FC with basal ganglia regions in schizophrenia subjects during a working memory task [11]. In resting-state fMRI increased functional integration in the caudate and decreased FC with the prefrontal and cerebellar regions was found in subjects with schizophrenia [9]. Interestingly, striatal connectivity indices have been used to identify treatment response in first episode psychosis subjects, with higher indices associated with non-responders and lower indices associated with responders [80], which supports the hypothesis that non-responders do not possess elevated striatal dopamine synthesis capacity [81]. These findings from the literature provide evidence that that our neuromechanistic biomarker is relevant in the pathophysiology of schizophrenia.

Third, using metastability as a single a-priori feature achieved classification performance in the range of previously published studies (see Table 6). Using the Cobre dataset, one study reported quite high levels of accuracy [82] in comparison to our study, and that of Morgan et al. [83]. However, it appears that the authors did not remove cases with significant framewise displacement which could explain the discrepancy.

Case positive indicates that either NAP or SCHZ was taken as the positive class for the classifier. Down-sampled indicates that the lack of balance between classes was rectified with random down-sampling. Random sampling indicates that a specific number of samples were

randomly chosen to allow balanced classes for external cross-validation. Blank cells indicate that the information was not available in the study manuscript.

When considering classification performance in different datasets, it appears that our classifier did not perform as well as the one from Morgan et al. [83] in the Dublin dataset, or with the one from Rashid et al. [24] in the Hartford dataset. However, in both cases the classes were not balanced (Dublin, cases:controls = 25:75, Hartford, cases:controls = 87:135) and there was no evidence that this was taken into consideration when reporting the performance, which may explain the discrepancy in the results.

We note that our cross-validation performance is comparable to the cross-validation performance in the other studies. Given these comparisons, we consider that our classifier had similar performance to those reported in the literature for cross-validation.

When we compare external validation of our classifier to that of Morgan et al. [83], we see that performance is similar with the same caveat pertaining to the Dublin dataset. We therefore consider that our cross-dataset analysis based on a single a-priori feature of metastability (as measured with VAR) performs similarly to one in the literature based on over 40'000 features in FC.

It is interesting to note that the HCPEP->Cobre external validation returned very high specificity whilst the Cobre->HCPEP external validation returned very high sensitivity. This may be related to the age difference between the participants in the two studies, or it may reflect that the cases in HCPEP are in the early stages of schizophrenia whilst the cases in Cobre are in a well-established stage of schizophrenia. It appears that disruptions in connectivity in early psychosis are not sufficient to distinguish SCHZ from CON. However, disruptions in SCHZ are sufficient to distinguish NAP from CON. This seems to imply that the disruptions in early psychosis are a subset of those in established schizophrenia.

Fourth, VAR explains between 81–92% of the variance in metrics for global integration, functional segregation, and the Metastability Index in both HCPEP RUN2 and Cobre datasets. Although metrics for integration, segregation, and conventional metastability have been estimated previously [78], the explanatory power of metastability on these global metrics of cerebral organization was not investigated. From our study we found that the conventional metric for metastability, META, explained between 0–0.08% of the variance in these metrics in the HCPEP RUN2 dataset, and between 0.16–0.55% in the Cobre data (see S3 Table). For HCPEP this is not surprising as no statistically significant difference was found in global metastability between CON and NAP in RUN2 (see S2 Data). For CON in Cobre, between 37–55% of the variance in the metrics was explained by metastability, but only 15–18% in SCHZ (see S3 Table). From these results it appears that antiphase synchrony is more prominent in SCHZ.

Our four complementary perspectives of group-level discrimination, individual-level classification, pathophysiological relevance, and explanatory power, provide preliminary evidence for the face validation of metastability, as measured with VAR, as a candidate neuromechanistic biomarker of schizophrenia.

There are several limitations that should be considered when evaluating the findings from this study. First, we used a novel proxy for metastability. Although the concept of metastability is generally accepted, its operationalization takes a number of forms from the entropy of spectral density [36], to the variability in spatial coherence [84], and to the most commonly used form, the standard deviation of the Kuramoto order parameter (phase synchrony) [67]. We chose the mean variance of instantaneous phase-locking as an alternative proxy for metastability based on the theory of Synergetics [50] and recent generalization of the Haken-Kelso-Bunz (HKB) model to multiple oscillators [51], which exhibits stable antiphase synchronization [52]. See Fig 10 reproduced with permission from [52]. It should be noted that the generalized

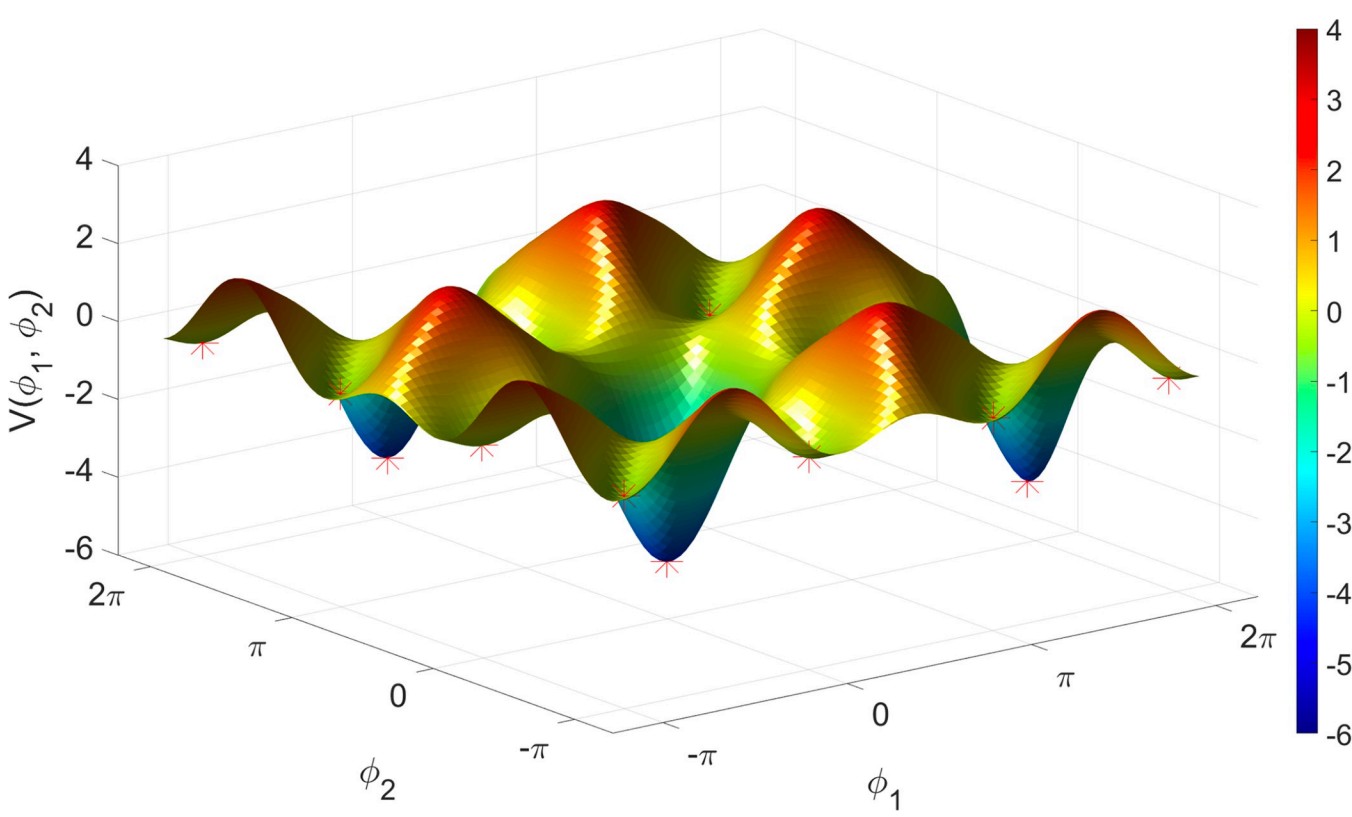

**Fig 10. Attractor landscape for the extended HKB model of multi-adic coordination.** A plot of the relative phase potential function landscape for $A_{ij} = 2B_{ij} = 1$ for each i, j. Note the many valleys (marked with red asterisks) in which an oscillator moving around in this landscape will become trapped. These valleys are the local minima corresponding to the coordination states. There are two types of valleys in this landscape: in-phase valleys, which have relatively very deep and wide basins of attraction, and antiphase valleys, which are narrower and shallower, reflecting the fact that the in-phase state is more stable than the antiphase state. Each of these valleys is separated by a distance of $\Pi$, and repeats infinitely on the potential surface in a $2\Pi$-periodic pattern. A, B, effective coupling parameters; i,j, i[th] and j[th] oscillator. Reproduced with permission from [52].

HKB model reduces to the Kuramoto model when second-order coupling is removed (i.e., $B_{ij} = 0$), and so can be seen as an extension of the Kuramoto model as in [85–87].

The generalized HKB model may explain the phase-locking behavior we illustrated in Fig 1 including mono-stability, bi-stability, switching ('sans switch') [88], and chimeras [67]. We have used a phenomenological understanding of the generalized HKB model to propose the mean variation of instantaneous phase-locking as a new proxy measure for metastability. In future work we need to perform a more thorough theoretical investigation of the phenomenon of metastability, complemented with a computational model that predicts empirical findings.

Second, from the perspective of alternative dFC approaches and pipelines, we did not perform global signal, white matter or cerebral spinal fluid regression. From a complexity science perspective [39], one cannot explore any subsystem of a complex system such as the brain in isolation, and accumulating evidence points to contributions other than neuronal to the fMRI signal [89–91]. As in [47], we defined communities of oscillators directly from the phase-locking data and not from intrinsic connectivity networks [41] nor predefined templates [68]. This allows regions to participate in multiple communities reflecting transient coalitions between the regions as evidenced by spatial overlap between networks [92, 93].

Third, the switching behavior observed in the phase-locking behavior in Fig 1 may appear to be artifactual. In LEiDA the leading eigenvector time-series is smoothed through a technique called "half-switching". We reproduced the time-series for one subject without this

smoothing and compared the results to the smoothed version. As may be seen in S5 Fig switching also occurs in the non-smoothed version, but with higher frequency than in the smoothed version.

However, since this smoothing was applied to all subjects, it does not affect the results, but may impact the ability to compare results with those obtained with alternative dFC approaches.

## Conclusion

This study claims face validity of metastability as a candidate neuromechanistic biomarker of schizophrenia based on group-level discrimination, individual-level classification, pathophysiological relevance, and explanatory power, congruent with published literature. Replication studies with larger sample sizes, method validation, and biomarker qualification need to be performed before claiming metastability to be a biomarker for clinical use. While diagnostic biomarkers of schizophrenia—such as metastability—may still have limited clinical utility, they can provide mechanistic insights for the discovery of prognostic biomarkers that could support treatment decisions. For example, the ability to identify treatment resistance or transition likelihood from high risk to first episode psychosis would address a real clinical need. Developing a deeper understanding of metastability may one day help us to gain sufficient mechanistic insight into the disconnection phenomenon of schizophrenia, which may lead in turn into the development of such effective biomarkers.

## Materials and methods

### Participants

**HCPEP.** Healthy controls (CON, $n$ = 53) and non-affective psychosis (NAP, $n$ = 82) participants were scanned at one of four sites (Indiana University, Beth Israel Deaconess Medical Center–Massachusetts Mental Health Center, McLean Hospital and Massachusetts General Hospital) as part of the Human Connectome Project-Early Psychosis (Principal Investigators: Shenton, Martha; Breier, Alan; U01MH109977-01, HCP-EP;

doi:10.15154/1524263 https://nda.nih.gov/edit_collection.html?id=2914) with funding from the National Institute of Mental Health (NIMH). A Data Use Certification (DUC) is required to access the HCPEP on the NIMH Data Archive (NDA).

NAP participants met DSM-5 criteria for schizophrenia, schizophreniform, schizoaffective, psychosis NOS, delusional disorder, or brief psychotic disorder with onset within the past five years prior to study entry. Additional inclusion/exclusion criteria may be found in https://www.humanconnectome.org/storage/app/media/documentation/data_release/HCP-EP_Release_1.0_Manual.pdf. See Table 7 for group demographics.

Procedures were approved by the Partners Healthcare Human Research Committee/IRB and complied with the Declaration of Helsinki. Participants provided written informed consent, or in the case of minors, parental written consent and participant assent.

**Cobre.** Neuroimaging data was obtained from the publicly available repository Cobre (http://fcon_1000.projects.nitrc.org/indi/retro/cobre.html) preprocessed with NIAK 0.17—lightweight release [94, 95]. The neuroimaging data included preprocessed resting-state fMRI data from healthy controls (CON, $n$ = 72) and schizophrenia patients (SCHZ, $n$ = 72), in which participants passively stared at a fixation cross. Subject recruitment and evaluation may be found in [96]. The study was approved by the institutional review board (IRB) of the University of New Mexico (UNM) and all subjects provided written informed consent. Inspection of the fMRI data for each subject resulted in the exclusion of one subject whose data did not include all 150 volumes. 13 SCHZ subjects with framewise displacement > 0.7mm were also

**Table 7. Demographic characteristics of participant groups.**

| Characteristics | HCP EP | | | COBRE | | |
|---|---|---|---|---|---|---|
| | HCs (n = 53) | NAPs (n = 82) | *p* | HCs (n = 71) | SCHZs (n = 59) | *p* |
| Age (years) | 24.85 ± 4.15 | 23.42 ± 3.57 | <0.001 | 35.88 ± 11.74 | 37.89 ± 13.86 | 0.39 |
| Sex (male/female) | 34/19 | 55/27 | 0.78 | 50/21 | 49/10 | 0.15 |
| Site (IU/BIDMC/MGH/MH) | 25/5/10/13 | 51/14/5/12 | | | | |
| PANSS | | 53.42 ± 10.02 | | | 48.67 ± 13.75 | |
| CAINS | | 21.48 ± 10.07 | | | | |
| IQ (WASI-II) (130 subjects recorded) | 116.30 ± 10.96 | 97.52 ± 17.55 | | 113.64 ± 12.57 | 102.82 ± 16.72 | |

HCP EP = Human Connectome Project for Early Psychosis; HC = Healthy Controls; NAP = patients with non-affective psychosis; SCHZ = patients with schizophrenia; PANSS = Positive and Negative Syndrom Scale; CAINS = Clinical Assessment Interview for Negative Symptoms; IQ = intelligence quotient. Significance test used for Age: Wilcoxon, Sex: Chi-squared

removed. The final dataset therefore used for the Cobre analysis included *n* = 59 SCHZ cases and *n* = 71 HCs. See Table 7 for group demographics and significance of between-group differences in age and sex.

## Image acquisition—HCPEP

All MRI scans were acquired on Siemens MAGNETOM Prisma 3T scanners with a multiband acceleration factor of 8, and a 32/64channel head coil. Each participant underwent four scans of resting-state fMRI collected over two experimental sessions on consecutive days (two scans in each session). The four datasets are referred to as RUN1 to RUN4. During each scan 410 frames were acquired using a multiband sequence at 2 mm isotropic resolution with repetition time (TR) of 0.72 sec over the space of 4 min 55 secs. The two scans in each session differed only in the phase encoding direction of anterior-posterior (AP) followed by posterior-anterior (PA) on both days.

## Image acquisition—Cobre

The resting-state fMRI data featured 150 echo planar imaging volumes obtained in 5 min, with repetition time (TR) = 2 s, echo time = 29 ms, acquisition matrix = 64×64 mm$^2$, flip angle = 75° and voxel size = 3×3×4 mm$^3$. The acquisition is fully described in detail in [96].

## Preprocessing

**HCPEP.** Data were pre-processed with the HCP's minimal pre-processing pipeline, and denoising was performed by the ICA-FIX procedure [97–99]. A complete description of the pre-processing details may be found at the HCP website https://www.humanconnectome.org/software/hcp-mr-pipelines. Briefly, fMRI data was gradient-nonlinearity distortion corrected, rigidly realigned to adjust for motion, fieldmap corrected, aligned to the structural images, and then registered to MNI space with the nonlinear warping calculated from the structural images. ICA-FIX was then applied on the data to identify and remove motion and other artifacts in the time-series.

**Cobre.** The preprocessing of the fMRI data is fully described in detail in [95, 96].

In brief, preprocessing included slice-timing correction, co-registration to the Montreal Neurological Institute (MNI) template and resampling of the functional volumes in the MNI space at a 6 mm isotropic resolution. We resampled the functional volumes in MNI space at a 2 mm isotropic resolution with 3dresample from AFNI [100]. Covariate removal was not

performed as subjects with excessive movement were removed and the time-series were later filtered between 0.01–0.08 Hz to remove low frequency drift and high frequency noise.

Substantial material in the following subsections is recycled from our prior publication [47].

## Parcellation

We parcellated the pre-processed fMRI data by averaging time-courses across all voxels for each region defined in the anatomical parcellation AAL [53] considering all cortical, subcortical, and cerebellar regions, $N$ = 116. We chose the AAL parcellation as subcortical and cerebellar regions are relevant in studies with psychiatric cohorts [3, 9, 101–103].

## Bandpass filtering

To isolate low-frequency resting-state signal fluctuations, we bandpass filtered the parcellated fMRI time-series within 0.01–0.08 Hz with a discrete Fourier transform (DST) computed using a fast Fourier transform (FFT) algorithm in MATLAB 2021b. We applied Carson's empirical rule [104, 105] on the analytical signal which was calculated using the Hilbert transform of the real signal [106], to confirm non-violation of the Bedrosian theorem for our bandpassed signals in both datasets (see S3 and S4 Figs).

## Functional connectivity through phase-locking

We estimated functional connectivity (FC) with the nonlinear measure of phase-locking which may be more suitable than linear measures such as Pearson correlation for analyzing complex brain dynamics. Specifically, nonlinear methods provide insight into interdependence between brain regions at both short and large time and spatial scales allowing the analysis of complex nonlinear interactions across space and time [107, 108]. From a practical perspective, unlike correlation or covariance measures, phase synchronization can be estimated at the instantaneous level and does not require time-windowing. When averaged over a sufficiently long-time window, phase-locking values provide a close approximation to Pearson correlation, varying within the same range of values [54, 109].

Following [54] we first calculated the analytical signal using the Hilbert transform of the real signal [106]. Then, the instantaneous phase-locking between each pair of brain regions $n$ and $p$ was estimated for each time-point $t$ as the cosine difference of the relative phase as

$$iPL(n, p, t) = cos(\theta(n, t) - \theta(p, t)) \tag{1}$$

Phase-locking at a given timepoint ranges between -1 (regions in antiphase) and 1 (regions in-phase). For each subject the resulting $iPL$ was a three-dimensional tensor of size $NxNxT$ where $N$ is the dimension of the parcellation, and $T$ is the number of timepoints in the scan.

## LEiDA–Leading Eigenvector Dynamic Analysis

To reduce the dimensionality of the phase-locking space for our dynamic analysis, we employed the Leading Eigenvector Dynamic Analysis (LEiDA) [54] method. The leading eigenvector $V_1(t)$ of each $iPL(t)$ is the eigenvector with the largest magnitude eigenvalue and reflects the dominant FC (through phase-locking) pattern at time $t$. $V_1(t)$ is a $Nx1$ vector that captures the main orientation of the fMRI signal phases over all anatomical areas. Each element in $V_1(t)$ represents the projection of the fMRI phase in each region into the leading eigenvector. When all elements of $V_1(t)$ have the same sign, this means that all fMRI phases are oriented in the same direction as $V_1(t)$ indicating a global mode governing all fMRI signals.

When the elements of $V_1(t)$ have both positive and negative signs, this means that the fMRI signals have different orientations behaving like opposite anti-nodes in a standing wave. This allows us to separate the brain regions into two 'communities' (or poles) according to their orientation or sign, where the magnitude of each element in $V_1(t)$ indicates the strength of belonging to that community [110]. For more details and graphical representation see [55, 57, 58]. The outer product of $V_1(t)$ reveals the FC matrix associated with the leading eigenvector at time $t$.

## Mode extraction

To identify recurring spatiotemporal modes $\psi$ or phase-locking patterns, we clustered the leading eigenvectors for each of the 10 phase-locked time-series datasets (HCPEP:CON x 4 runs, HCPEP:NAP x 4 runs, Cobre:CON x 1 run, Cobre:SCHZ x 1 run) with K-means clustering with 300 replications and up to 400 iterations for 2–10 centroids. This approach is similar to a previous study [47] but different from other studies that used LEiDA where k-means clustering was either performed on concatenated datasets across groups [54–56], or where the centroids extracted from one group were used to cluster other groups [57–59]. This approach considers each dataset as a unique observation of brain activity with associated variability in the spatiotemporal modes and avoids data leakage [60]. K-means clustering returns a set of K central vectors or centroids in the form of $Nx1$ vectors $V_c$. As $V_c$ is a mean derived variable, it may not occur in any individual subject data set. To obtain time courses related to the extracted modes $\psi_k$ at each TR we assign the cluster number to which $V_c(t)$ is most similar using the cosine distance.

## Mode representation in voxel space

To obtain a visualization in voxel space of the spatial modes $V_c$ we first reduced the spatial resolution of all fMRI volumes from 2mm$^3$ to 10mm$^3$ to obtain a reduced number of brain voxels (here $N = 1821$) to be able to compute the eigenvectors of the $NxN$ phase-locking matrices. The analytic signal of each 10mm$^3$ voxel was computed using the Hilbert transform, and the leading eigenvectors were obtained at each time point (with size $NxT$). Subsequently, the eigenvectors were averaged across all time instances assigned to a particular cluster, obtaining in this way, for each cluster, a $1xN$ vector representative of the mean phase-locking pattern captured in voxel space.

## Mode representations as connectograms

We visualized FC as connectograms by taking the FC matrices for each mode and retaining regions that were collectively in-phase but in antiphase with the global mode.

## Neurosynth functional associations

Probabilistic measures of the association between brain coordinates and overlapping terms from the Cognitive Atlas [111] and the Neurosynth database [72] were obtained as in [73]. The probabilistic measures were parcellated into 116 AAL regions and may be interpreted as a quantitative representation of how regional fluctuations in phase-locking are related to psychological processes. The resulting functional association matrix represents the functional relatedness of 130 terms to 116 brain regions (see S4 Table for a full list of terms).

## Metastability

Empirical metastability studies to date have used pre-defined resting-state networks (RSN) extracted with ICA [41], with network masks [68], or with functional templates [112] to

represent communities of oscillators for investigation of network synchrony and metastability. The non-overlapping nature of these networks does not allow flexible allegiance of brain regions to different communities [69]. In contrast, as in [47] we decided to take a purely data driven approach, using the recurrent modes extracted with K-means clustering to represent communities of oscillators. As we decided to retain 5 recurrent modes (see Results), we therefore have 5 communities of oscillators $\psi_1-\psi_5$. Note that the AAL regions are not constrained to a single community and so the communities reflect time-varying coalitions among regions. The number of brain regions in each community for each of the 10 datasets is shown in S1 Table.

**Based on phase synchrony.** The Kuramoto order parameter in each community $\psi$ of $m$ regions is given by

$$R_\psi(t) = |\langle e^{i\theta(m,t)}\rangle|,\, _{m\in\psi} \tag{2}$$

Metastability was calculated as the standard deviation over time of the Kuramoto order parameter in each community. The mean value of this measure across communities denoted as global metastability, represents the overall variability in phase synchrony across communities.

If we fix time $t$ and estimate the variance of $R_\psi(t)$ across all communities $\psi_{1...5}$, we obtain an instantaneous measure of how chimera-like the system is at time $t$.

$$CHI(t) = var(|\langle e^{i\theta(\psi,t)}\rangle|),\, _{\psi_{1...5}} \tag{3}$$

where *CHI* is a measure of chimerality, an indicator of cluster synchronization [67]

**Based on phase-locking.** The instantaneous phase-locking (*iPL*) between each pair of brain regions $n$ and $p$ was estimated for each time-point $t$ as in Eq 1. Metastability, denoted as VAR to distinguish it from metastability above, was calculated as the mean of the variance of instantaneous phase-locking over time in each community. The mean value of this measure across communities denoted as global VAR represents the overall variance in the phase-locking across communities.

## Integration

Global integration was assessed as the connectivity within the time-averaged phase-locking matrix calculated with Eq 1. The matrix is scanned through all possible thresholds from 0 to 1, binarized, and the size of the largest connected component is identified using the Brain Connectivity Toolbox [113]. The integral of the size of the largest connected component as a function of the threshold is taken as an estimate of global integration [78].

## Segregation

Segregation refers to the decomposition of a system into functional subcomponents and was estimated with the modularity index Q of the time-average phase locking matrix calculated with Eq 1. The Louvain algorithm was used to subdivide the matrix into modules with the Newman modularity Q taken as an estimate of functional segregations [78].

## Statistical analysis

**Interclass correlation coefficient.** ICC is a relative metric that is used for test-retest reliability in measurement theory [114]. It is generally defined as the proportion of the total measured variance that can be attributed to within subject variation. As such, ICC coefficients may

be low when there is little variance between subjects, that is in a homogeneous sample, or when the within-subject variance is large [115]. In this study we use the ICC forms from [116].

There are many scales for ICC, so for clarity we will use those of Landis and Koch [117]:

low ($0 < \text{ICC} < 0.2$)

fair ($0.2 < \text{ICC} < 0.4$)

moderate ($0.4 < \text{ICC} < 0.6$

substantial ($0.6 < \text{ICC} < 0.8$)

almost perfect ($0.8 < \text{ICC} < 1$)

We calculated the run reliability of mode $\psi$ extraction with ICC(1,1) in search of agreement rather than consistency across runs [64].

**Parametric testing.** Before performing statistical tests, we checked if the assumptions for parametric testing were met. In all cases, the assumptions were violated. The results of these tests can be found for basal ganglia in S5 Table, global metastability in S6 Table, local metastability in S7 Table, global VAR in S8 Table, local VAR in S2 Table.

**Non-parametric ANOVA testing.** We used Align rank transform (ART) [65, 66] to perform multi-factor non-parametric testing with dependent groups in R (ARtool::art). We then followed the statistical testing flowchart shown in Fig 11. All results were Bonferroni corrected for multiple comparisons.

**Non-parametric permutation t-tests.** We used permutation Welch 2 sample t-tests with $n = 9999$ Monte Carlo permutations implemented in R (MKinfer::perm.t.test) as the majority of distributions were not normally distributed when assessed with a Shapiro test.

**Classification of condition based on metastability.** Supervised machine learning algorithms were trained to classify cases and controls for each dataset independently using a single a-priori feature of metastability as measured by VAR. Classification was performed using a naïve Bayes non-linear classification model [118] in R implemented with Caret [119]. We used a naïve Bayes classifier as we had just one feature with no issue of independence. For the HCPEP datasets, we chose cross-validation over internal validation in a different run to avoid data leakage, as the same participants would have been present in both the test and validation sets [60].

In all five datasets (4 HCPEP datasets, 1 Cobre dataset), we assessed the generalizability of the classifier using repeated k-fold cross-validation, $k = 10$, *repetitions* = 20. For the out-of-sample analysis we trained the classifier in HCPEP and tested it in Cobre; and trained the classifier in Cobre and tested it in HCPEP. For all datasets we used down-sampling to balance the classes, and for the out-of-sample analysis we randomly down-sampled both datasets to 53 to allow cross-dataset testing. We report the area under the operating characteristics curve (AUC), balanced accuracy, sensitivity, and specificity. The statistical significance of balanced accuracy was assessed with a binomial cumulative distribution function [60].

**Software tools.** Parcellation, LEiDA, ICC and metastability / VAR derivations were implemented in MATLAB [120]. Neurosynth functional associations were derived in Python 3.8.5. All other statistical analysis were performed in RStudio Team version 2022.02.3 Build 492 [121].

## Supporting information

**S1 Fig. Silhouette values for clustering solutions for 1 to 9 clusters with 2–10 modes respectively.** (**A**) HCPEP CON. (**B**) HCPEP NAP. (**C**) Cobre CON (**D**) Cobre SCHZ.
(TIF)

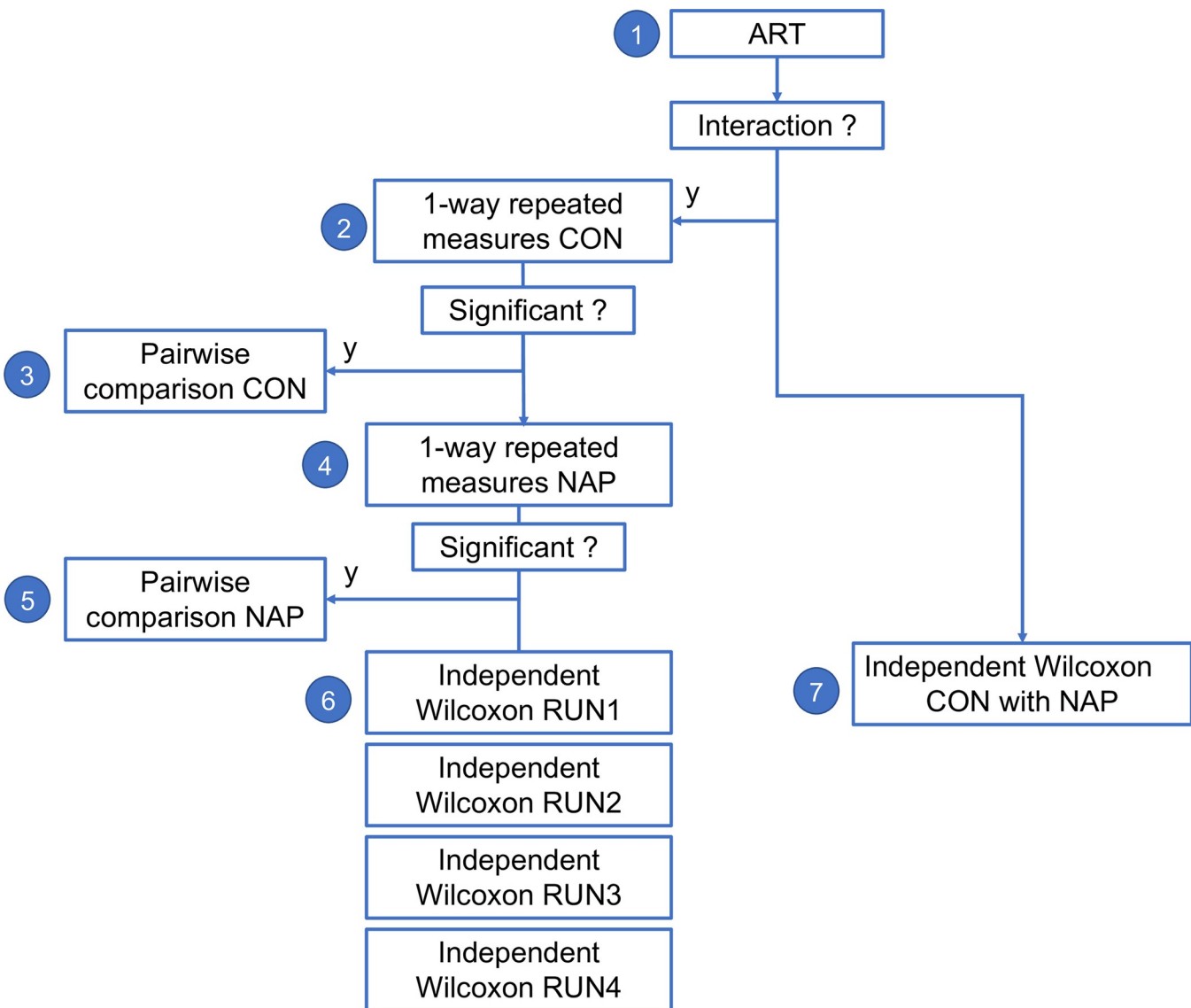

**Fig 11. Statistical flowchart for non-parametric testing of differences between groups across runs.** 1) 2x4 non-parametric ANOVA using Align rank transform (ART). 2) Friedmann repeated measures test. 3) Paired Wilcoxon test. 4) Friedmann repeated measures test. 5) Paired Wilcoxon test. 6) Independent Wilcoxon test for each run. 7) Independent Wilcoxon test across all runs.

**S2 Fig. Reliability of mode extraction for controls and non-affective psychosis.**
(TIF)

**S3 Fig. Non-violation of Bedrosian Theorem–HCPEP.**
(TIF)

**S4 Fig. Non-violation of Bedrosian Theorem–Cobre.**
(TIF)

**S5 Fig. Effect of smoothing on the leading eigenvector time-series.** A) Time-series for the leading eigenvectors for one subject without smoothing. B) Time-series for the leading eigenvector for the same subject with half-switch smoothing. The blue asterixis indicate that half-

switching occurred.
(TIF)

**S1 Table. Number of brain regions in each community across datasets.** We used the spatio-temporal modes to define communities for estimation of metastability.
(XLSX)

**S2 Table. Assumption test results for mode VAR in the HCPEP and Cobre datasets.** We assessed the normality of the distribution of VAR in each mode with a Shapiro-Wilk test, equivalence of variance with a Levine test, and effect size with Cohen's D test.
(XLSX)

**S3 Table. Linear regression statistics for global integration, functional segregation, Metastability Index on META and VAR.** We performed linear regression in HCPEP RUN2 and Cobre for 3 metrics of integration on both META and VAR.
(XLSX)

**S4 Table. Neurosynth terms.**
(XLSX)

**S5 Table. Assumption test results for contribution of basal ganglia regions FC in the HCPEP dataset.** We assessed the normality of the distribution of contribution with a Shapiro-Wilk test, equivalence of variance with a Levine test, and effect size with Cohen's D test.
(XLSX)

**S6 Table. Assumption test results for global META in the HCPEP and Cobre datasets.** We assessed the normality of the distribution of META with a Shapiro-Wilk test, equivalence of variance with a Levine test, and effect size with Cohen's D test.
(XLSX)

**S7 Table. Assumption test results for metastability in the modes in the HCPEP and Cobre datasets.** We assessed the normality of the distribution of mode META with a Shapiro-Wilk test, equivalence of variance with a Levine test, and effect size with Cohen's D test.
(XLSX)

**S8 Table. Assumption test results for global VAR in the HCPEP and Cobre datasets.** We assessed the normality of the distribution of META with a Shapiro-Wilk test, equivalence of variance with a Levine test, and effect size with Cohen's D test.
(XLSX)

**S1 Data. Results from statistical tests for differences in basal ganglia connectivity between HC and NAP in the HCPEP dataset.**
(XLSX)

**S2 Data. Results from statistical tests for differences in mode META between HC and NAP in the HCPEP dataset, and HC and SCHZ in the Cobre dataset.**
(XLSX)

**S3 Data. Results from statistical tests for differences in mode VAR between HC and NAP in the HCPEP dataset, and HC and SCHZ in the Cobre dataset.**
(XLSX)

**S1 Text. Analysis of in-phase synchrony based Metastability.**
(PDF)

## Author Contributions

**Conceptualization:** Fran Hancock.

**Data curation:** Robert A. McCutcheon.

**Formal analysis:** Fran Hancock.

**Investigation:** Fran Hancock.

**Methodology:** Fran Hancock, Joana Cabral.

**Software:** Fran Hancock.

**Supervision:** Ottavia Dipasquale, Federico E. Turkheimer.

**Validation:** Fran Hancock.

**Visualization:** Fran Hancock.

**Writing – original draft:** Fran Hancock, Fernando E. Rosas, Robert A. McCutcheon.

**Writing – review & editing:** Fran Hancock, Fernando E. Rosas, Robert A. McCutcheon, Joana Cabral, Ottavia Dipasquale, Federico E. Turkheimer.

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
