## [Decision Letter · Decision Letter 0]

19 Dec 2022

PONE-D-22-28431Metastability as a neuromechanistic biomarker of schizophrenia pathologyPLOS ONE

Dear Dr. Hancock,

Thank you for submitting your manuscript to PLOS ONE. After careful consideration, we feel that it has merit but does not fully meet PLOS ONE’s publication criteria as it currently stands. Therefore, we invite you to submit a revised version of the manuscript that addresses the points raised during the review process.

We look forward to receiving your revised manuscript.

Kind regards,

Daqing Guo

Academic Editor

PLOS ONE

Journal Requirements:

4. Thank you for stating the following in the Competing Interests/Financial Disclosure * (delete as necessary) section:

“RM 200/02/Z/15/Z Wellcome Trust Career Development fellowship https://wellcome.org/grant-funding/schemes/career-development-awards

JC UIDB/50026/2020, UIDP/50026/2020 and CEECIND/03325/2017 Portuguese Foundation for Science and Technology https://3bs.uminho.pt/research-projects

FET National Institute for Health Research (NIHR) Biomedical Research Centre (BRC) at South London and Maudsley NHS Foundation Trust and King’s College London 

OD National Institute for Health Research (NIHR) Biomedical Research Centre (BRC) at South London and Maudsley NHS Foundation Trust and King’s College London 

The funders had no role in the study design, data collection and analysis, decision to publish, or preparation of the manuscript.”

We note that you received funding from a commercial source:  “Wellcome Trust”

6. Thank you for stating the following in the Competing Interests section:

“I have read the journal's policy and the authors of this manuscript have the following competing interests: RM has received honoraria for educational talks from Otsuka and Janssen”

Reviewers' comments:

Reviewer's Responses to Questions

**Comments to the Author**

1. Is the manuscript technically sound, and do the data support the conclusions?

Reviewer #1: Partly

Reviewer #2: Yes

Reviewer #3: Yes

2. Has the statistical analysis been performed appropriately and rigorously? 

Reviewer #1: Yes

Reviewer #2: Yes

Reviewer #3: Yes

3. Have the authors made all data underlying the findings in their manuscript fully available?

Reviewer #1: Yes

Reviewer #2: Yes

Reviewer #3: Yes

4. Is the manuscript presented in an intelligible fashion and written in standard English?

Reviewer #1: Yes

Reviewer #2: Yes

Reviewer #3: Yes

5. Review Comments to the Author

Reviewer #1: In this study, the authors applied Leading Eigenvector Dynamic Analysis (LEiDA) to capture specific features of dynamic functional connectivity and then implements a novel approach to estimate metastability. They found that the new approach was capable of discriminating cases from controls with elevated effect sizes relative to published literature, reflected in an up to 76% area under the curve (AUC) in out-of-sample classification analyses. Furthermore, their analyses showed that patients with early psychosis exhibit intermittent disconnectivity of subcortical regions with frontal cortex and cerebellar regions, introducing new insights about the mechanistic bases of these conditions. These results demonstrate reliability and face validity of metastability as a neuromechanistic biomarker of schizophrenia pathology. The paper is overall well-written. Nevertheless, there are still a few issues to be addressed.

Questions:

1) (Introduction p1) The authors only briefly introduced schizophrenia, please elaborate on the symptoms of schizophrenia, such as those related to the disconnection hypothesis, and the authors said that "Biomarkers of schizophrenia in early and established phases may differ, …", which also needs to be explained in detail.

2) (Introduction p4) I noted that the authors analyzed the suitability of a specific marker of brain dynamics: metastability, which is a ubiquitous concept across diverse models of brain functioning including coordination dynamics and complex systems. Please elaborate on the reasons for choosing it and its advantages over commonly used dynamic metrics.

3) (Method) Statistical methods should be used to examine whether there are between-group differences in the demographic characteristics of the participants.

4) (Method) Covariate removal was not performed in the preprocessing of Cobre data. Would this affect subsequent analyses?

5) I found some typographical and word errors. Please check the manuscript carefully. For example, ‘mm2, mm3’ and ‘… an ubiquitous’.

6) (Discussion) “…, This may reflect that the cases in HCPEP are in the early stages of schizophrenia whilst the cases in Cobre are in a well-established stage of schizophrenia.”

In fact, the age of participants in HCPEP and Cobre datasets was quite different, whether this would affect the validation between the two datasets.

Reviewer #2: Hancock and colleagues examined whether metastability derived from resting-state fMRI can be used as a mechanistic biomarker of schizophrenia, including a demonstration of reliability and face validity. To this end, using two datasets from the HCP Early Psychosis (HCPEP) and Cobre, they applied Leading Eigenvector Dynamic Analysis (LEiDA) technique to capture specific features of dynamic functional connectivity and then introduced a novel approach to estimate metastability. They achieved an AUC of 76% in the classification analysis and found patients with early psychosis show intermittent disconnectivity of subcortical regions with frontal cortex and cerebellar regions. The topic of the paper is of interest, but I have some suggestions to improve the paper’s likely reach and impact:

One concern I have is about the novelty of the methods presented. The authors claimed that it is a “new” approach, but one of the coauthors (Cabral and colleagues, 2017) has already published the very similar work. Her previous work introduced a series of steps (e.g., phase-locking based FC and iPL computation) to apply LEiDA approach to fMRI data. The cited paper by Cabral et al. already presents large parts of the current methods and results. My understanding is that the authors used the existing methods. So, it is not clear what is “new” approach compared to previous work by Cabral et al.. What is the novelty of this work in terms of methodology? Just difference in metastability as the mean variance of instantaneous phase-locking, VAR compared to the conventional metastability? What if we used the traditional metastability values in analyses? This metastability metric cannot be used as a biomarker of schizophrenia? Supporting evidence to claim it as a biomarker of SCZ should be provide in more detail. At current form, it sounds over-claimed statement/conclusion given a small number of samples (patients with SCZ and early psychosis).

Also, I encourage to compare the results by a “new” metastability and those by a “conventional” metastability in analyses, including experiments to provide the insight about a mechanistic explanation as well as classification ability.

Is there any difference in metastability due to the choice of resting-state networks? As described in the paper, one can choose pre-defined resting-state network extracted with ICA, network masks or functional template. First, why the authors chose data-driven approach rather than the existing ones? Second, the choice of the template to define different resting-state networks affected the classification ability?

As an additional concern, I noted that metastability for each network was computed as the mean value of the variance of instantaneous phase-locking over time in each community/network. Different communities have different number of brain ROIs included in analysis. Synchrony across a large number of ROIs in a community/network will have a smaller variance. In this sense, I am somewhat unsurprised that these networks demonstrate lower levels of both metastability and synchrony in comparison to other networks with smaller number of ROIs. This is a factor that should be included in the statistical analysis to ensure the robustness of this result. This should be discussed in the discussion.

In building machine learning classifier using metastability measures, first of all, I wondered why the authors chose a naïve Bayes model rather than other common models such as support vector machine or logistic regression. I am curious whether the authors considered other machine learning classifiers and tested to evaluate the performance of the binary classification (SCZ vs. controls). Second, were there any pre-processing step prior to building a machine learning classifier such as standardization or min-max scaling and so on. Third, when using a naïve Bayes model, how to tune the hyperparameters to obtain optimal parameters that may provide best performance in classification. Fourth, why doswnsampling? This may lead to overfitting due to smaller number of subjects in controls and SCZ. I was wondering if up-sampling approach to the samples has been considered to make the samples balanced.

In terms of evaluating the performance of the classifier, the authors used the HCPEP as a training set and the Cobre as a testing set and vice versa for the out-of-sample analysis. Table 4 shows the results of out-of-sample testing for both scenarios. Can you provide more detailed discussion on why the results are different due to the choice of which training sample was used. Overall, a classifier trained on Cobre sample provided better performance than the other case. Any discussion on these results? Further, given a balanced datasets downsampled, the results by classification are not good in terms of accuracy (0.38-0.57) and other measures as well. Why?

I like the way of testing the machine learning model by using different, independent samples. Have the authors ever tested the classification performance within each sample, although the sample size is quite small?

Reviewer #3: In this article, the preprocessed data of HCPEP and Cobre datasets were applied the Hilbert transform, and the authors introduced a new method of Leading Eigenvector Dynamic Analysis (LEiDA) in the frequency domain to capture the specific features of dynamic functional connectivity and then implemented a novel approach to estimate the metastability. Finally, the control group and cases group were classified by a naïve Bayes classifier.

Due to the unclear description of the method, there may be some problems with the results. Some comments are listed as follows:

1.There are several places where the data are not clear.

(1) The number of healthy controls and patients in the abstract is not consistent with that in the manuscript. (2) Line 817, page 36: Which five datasets do they refer to?

2.Line 702, page 31: “3 conditions x 4 runs” refers to unknown. If “3 conditions” refers to the healthy control group, non-affective psychosis, and schizophrenia respectively, image acquisition in the text only describes that there are 4 runs in the HCPEP dataset, while there are no 4 runs in Cobre dataset for the healthy control group and schizophrenia patients.

3.Line 732 indicates that S7 is included in the supplementary materials, but it is not.

4.The demographic characteristics of participant groups should include the subject information that meets the inclusion criteria of the experiment. In addition, statistical comparisons can be made on information such as gender and age of different groups to show whether there are significant differences between groups in terms of gender and age.

5.There are three minor problems.

(1) The full name of fMRI should appear once in the full text, such as the abstract.

(2) The full text does not indicate that iPL is the abbreviation of instantaneous phase-locking.

(3) There are some grammar problems in the article. Please check the grammar carefully. For example, it appears that you are missing a comma after the introductory phrase “In this study”. Consider adding a comma (Line 39, page 3). And it seems that the verb “implements” does not agree with the subject. Consider changing the verb form (Line 49, page 3).

6. PLOS authors have the option to publish the peer review history of their article (what does this mean?). If published, this will include your full peer review and any attached files.

Reviewer #1: No

Reviewer #2: No

Reviewer #3: No

---

## [Author Response · Author response to Decision Letter 0]

13 Jan 2023

Response to comments from reviewers of the paper

PONE-D-22-28431 

“Metastability as a neuromechanistic biomarker of schizophrenia pathology”

We would like to thank the editor and the reviewers for taking the time to read and think critically about our manuscript. The comments are encouraging, and the reviewers have raised several insightful questions and challenges. In response, we have undertaken additional analysis, and the findings add to the weight of evidence that metastability can indeed be considered a candidate neuromechanistic marker for schizophrenia pathology.

Please see below, in blue, our detailed point-to-point response to the comments. All page numbers refer to the manuscript file with tracked changes.

In this response:

Editor and reviewer comments are in black

Author responses are in green

Revised text from the manuscript is in blue with line numbers from the revised manuscript

Journal requirements

E1 We have made the appropriate corrections in the manuscript and accompanying files. 

E2/E3 We have addressed the question of data sharing in the revised cover letter, as directed.

E4/E5/E6 We have addressed the questions of financial disclosure and competing interests in the revised cover letter, as directed.

Reviewer #1

Reviewer #1: In this study, the authors applied Leading Eigenvector Dynamic Analysis (LEiDA) to capture specific features of dynamic functional connectivity and then implements a novel approach to estimate metastability. They found that the new approach was capable of discriminating cases from controls with elevated effect sizes relative to published literature, reflected in an up to 76% area under the curve (AUC) in out-of-sample classification analyses. Furthermore, their analyses showed that patients with early psychosis exhibit intermittent disconnectivity of subcortical regions with frontal cortex and cerebellar regions, introducing new insights about the mechanistic bases of these conditions. These results demonstrate reliability and face validity of metastability as a neuromechanistic biomarker of schizophrenia pathology. The paper is overall well-written. Nevertheless, there are still a few issues to be addressed.

Questions:

1.1) (Introduction p1) The authors only briefly introduced schizophrenia, please elaborate on the symptoms of schizophrenia, such as those related to the disconnection hypothesis, and the authors said that "Biomarkers of schizophrenia in early and established phases may differ, …", which also needs to be explained in detail.

Many thanks for the suggestions. We have included the symptoms of schizophrenia (lines 69-73) and related several of them to the disconnection hypothesis (lines 84-91). 

(lines 69-73)

Originally described as the fragmentation of previously integrated mental experiences [2], the disorder is associated with positive symptoms such as delusions, hallucinations, and disordered thoughts, negative symptoms including amotivation and social withdrawal, and cognitive symptoms including deficits in executive function [3].

(lines 84-91)

For example, auditory verbal hallucinations have been associated with aberrant coupling in the speech processing system, speech production system, and the auditor monitoring system [8]. Additionally, amotivation has been linked with aberrant connectivity between the caudate nucleus and the cerebellum, leading to impaired goal achievement behaviour, and with prefrontal areas leading to poor goal-directed performance [9]. Moreover, disorganized symptoms have been predicted by aberrant connectivity between the cerebellum and the cingulo-opercular and salience networks [10].

Additionally, we have created a section to explain differences in early psychosis and chronic schizophrenia to explain why biomarkers might differ during the course of illness (lines 97-111).

However, recent studies have highlighted differences in aberrant connectivity between early- and late-stage schizophrenia. Reduced cerebellum connectivity in early psychosis and increased connectivity in chronic schizophrenia were associated with both positive and negative symptom severity, suggesting a compensatory role for the cerebellum [15]. Disconnectivity between the somatosensory and visual networks was found to be pervasive in early psychosis, but not the disconnectivity between the default mode, cognitive control, and salience networks [16]. And finally, subcortical disconnectivity was found in early psychosis whilst both subcortical and cortico-subcortical disconnectivity was apparent in chronic schizophrenia. Importantly, the polarity of associations between disconnectivity and positive symptom severity were reversed for early and chronic groups, suggesting differences in neural correlates of psychotic symptoms at different stages of illness, and/or the potential effects of medication [17]. Hence, biomarkers of schizophrenia in early and established phases may differ, which may be informative of developing pathophysiology.

1.2) (Introduction p4) I noted that the authors analyzed the suitability of a specific marker of brain dynamics: metastability, which is a ubiquitous concept across diverse models of brain functioning including coordination dynamics and complex systems. Please elaborate on the reasons for choosing it and its advantages over commonly used dynamic metrics.

Following the reviewer’s suggestion, we have now elaborated on the reasons for choosing metastability over other commonly used markers (lines 134-147).

There are several reasons for choosing metastability as a marker of brain dynamics in schizophrenia. First, it reflects the competitive tension between integration and segregation, and is therefore relevant for studies based on the disconnection hypothesis. Second, commonly used dynamic metrics including group-level dwell/duration and occurrence/occupancy were found to differ significantly across multiple scanning sessions in healthy young adults, which could potentially blur state with trait variability [47]. Additionally, in that study, the reconfiguration process was found to be non-Gaussian, that is, there was memory in the network reconfiguration process. As such, it is not possible to use first-order Markov methods to calculate state transition probabilities. Moreover, the study concluded that global metastability was the only representative and stable metric of the 9 dynamic metrics investigated, highlighting its potential as a group-level biomarker of psychiatric disorders [47]. 

1.3) (Method) Statistical methods should be used to examine whether there are between-group differences in the demographic characteristics of the participants.

We have now performed Wilcoxon t-tests on age and Chi-squared tests on sex to identify between-group differences. The results are found in Table 7, line 789.

1.4) (Method) Covariate removal was not performed in the preprocessing of Cobre data. Would this affect subsequent analyses?

We have commented on the non-removal of covariate (lines 832-835).

Covariate removal was not performed as subjects with excessive movement were removed and the time-series were later filtered between 0.01-0.08 Hz to remove low frequency drift and high frequency noise.

1.5) I found some typographical and word errors. Please check the manuscript carefully. For example, ‘mm2, mm3’ and ‘… an ubiquitous’.

These errors have been corrected.

1.6) (Discussion) “…, This may reflect that the cases in HCPEP are in the early stages of schizophrenia whilst the cases in Cobre are in a well-established stage of schizophrenia.”

In fact, the age of participants in HCPEP and Cobre datasets was quite different, whether this would affect the validation between the two datasets.

We thank the reviewer for raising this point. We have noted that these differences could also be due to the age differences between the participants (lines 652-653).

This may be related to the age difference between the participants in the two studies, or it may reflect that the cases in HCPEP are in the early stages of

Reviewer #2

Reviewer #2: Hancock and colleagues examined whether metastability derived from resting-state fMRI can be used as a mechanistic biomarker of schizophrenia, including a demonstration of reliability and face validity. To this end, using two datasets from the HCP Early Psychosis (HCPEP) and Cobre, they applied Leading Eigenvector Dynamic Analysis (LEiDA) technique to capture specific features of dynamic functional connectivity and then introduced a novel approach to estimate metastability. They achieved an AUC of 76% in the classification analysis and found patients with early psychosis show intermittent disconnectivity of subcortical regions with frontal cortex and cerebellar regions. The topic of the paper is of interest, but I have some suggestions to improve the paper’s likely reach and impact:

2.1 One concern I have is about the novelty of the methods presented. The authors claimed that it is a “new” approach, but one of the coauthors (Cabral and colleagues, 2017) has already published the very similar work. Her previous work introduced a series of steps (e.g., phase-locking based FC and iPL computation) to apply LEiDA approach to fMRI data. The cited paper by Cabral et al. already presents large parts of the current methods and results. My understanding is that the authors used the existing methods. So, it is not clear what is “new” approach compared to previous work by Cabral et al.. What is the novelty of this work in terms of methodology? Just difference in metastability as the mean variance of instantaneous phase-locking, VAR compared to the conventional metastability?

We have changed the text in the abstract to make it clear that this work extends LEiDA (line 49) and elaborated on the novel aspects of the approach in both the results section (lines 190-197), (301-304), (314-318) and the methods section (900-905), (940-941).

(line 49)

In this work we extend Leading Eigenvector Dynamic Analysis (LEiDA)

(lines 190-197)

This is similar to a previous study [47], but different from other studies that used LEiDA where k-means clustering was either performed on concatenated datasets across groups [54–56] or where the centroids extracted from one group were used to seed the clustering of other groups [57–59]. The approach in this study considers each dataset as a unique observation of brain activity with associated variability in the spatiotemporal modes and avoids data leakage which occurs when dimensionality reduction is performed on the dataset as a whole [60].

(lines 301-304)

In contrast to previous studies of metastability in fMRI we choose not to use predefined templates [68], or intrinsic connectivity networks [41] to define our communities. The non-overlapping nature of these networks does not allow flexible allegiance of brain regions to different communities [69].

(lines 314-318)

in-phase synchrony and ignored antiphase synchrony. Whilst this may seem to be a small methodological difference, it in fact highlights conceptual differences in the understanding of mechanisms of co-ordination across the brain [70] and the role of antiphase synchrony in large-scale cortical networks [71].

(Lines 900-905) 

This approach is similar to a previous study [47] but different from other studies that used LEiDA where k-means clustering was either performed on concatenated datasets across groups [54–56], or where the centroids extracted from one group were used to cluster other groups [57–59]. This approach considers each dataset as a unique observation of brain activity with associated variability in the spatiotemporal modes and avoids data leakage [60].

(Lines 940-941).

The non-overlapping nature of these networks does not allow flexible allegiance of brain regions to different communities [69]. 

2.2 What if we used the traditional metastability values in analyses? This metastability metric cannot be used as a biomarker of schizophrenia? 

We answer this question under 2.15 and 2.4 below.

2.6 Second, the choice of the template to define different resting-state networks affected the classification ability?

We answer this question under 2.15 below.

2.8 In building machine learning classifier using metastability measures, first of all, I wondered why the authors chose a naïve Bayes model rather than other common models such as support vector machine or logistic regression. I am curious whether the authors considered other machine learning classifiers and tested to evaluate the performance of the binary classification (SCZ vs. controls).

We have explained our rational for choosing a naïve Bayes classifier (line 434-435) and answer the question on alternative classifiers in 2.15 below. 

Briefly, we chose a naïve Bayes classifier for its simplicity, but still compared its performance with other classifiers (as discussed below).

2.11 Fourth, why downsampling? This may lead to overfitting due to smaller number of subjects in controls and SCZ. I was wondering if up-sampling approach to the samples has been considered to make the samples balanced. 

We answer this question under 2.15 below.

2.15 I like the way of testing the machine learning model by using different, independent samples. Have the authors ever tested the classification performance within each sample, although the sample size is quite small?

We performed classification of subjects using the conventional metric for metastability. The performance is significantly poorer than with VAR. Additionally, addressing comments 2.2, 2.6, 2.8, 2.11, and 2.15, we ran the classifier with up-sampling, logistic regression, SVM, an intrinsic connectivity template, NeuroMark, and with internal validation in RUN1 for HCPEP. The results are compared in Table 5 (line 485) and additional text has been added to the section (lines 477-483, 488-492).

(lines 477-483)

In addition to a down-sampled naïve Bayes classifier, we also considered an up-sampled naïve Bayes, down-sampled Logistic regression, and a down-sampled Support Vector Machine models for VAR in mode ψ_4. Additionally, we used a down-sampled naïve Bayes classifier for META in mode ψ_4, for VAR in mode ψ_4 when calculated using NeuroMark [74] intrinsic connectivity networks, and for internal validation for HCPEP trained in RUN2 and tested in RUN1. The performance of these additional classifications is shown in Table 5.

Table 5 Performance of additional classifiers in comparison to the classifier used in this study. 

(lines 488-492)

As can be seen from Table 5, there was little difference in performance between down-sampling and up-sampling naïve Bayes, logistic regression, and Support Vector Machine classifiers. However, performance dropped significantly when the conventional metric for metastability, or a non-overlapping intrinsic connectivity network template, NeuroMark, were used in the classifier.

2.3 Supporting evidence to claim it as a biomarker of SCZ should be provided in more detail. At current form, it sounds over-claimed statement/conclusion given a small number of samples (patients with SCZ and early psychosis).

We have qualified the biomarker as a ‘candidate biomarker’ in the title and throughout the text, and acknowledged in the conclusion that replication studies, method validation, and biomarker qualification would be required before claiming it to be a biomarker for clinical use (lines 737-739).

Replication studies with larger sample sizes, method validation, and biomarker qualification need to be performed before claiming metastability to be a biomarker for clinical use.

2.4 Also, I encourage to compare the results by a “new” metastability and those by a “conventional” metastability in analyses, including experiments to provide the insight about a mechanistic explanation as well as classification ability.

We extended the analysis to include a measure of global integration, functional segregation, and a metastability index, and investigate their relationship with META and VAR. We have created a new section (lines 494-563), and based on the results, updated the Discussion section (lines 660-677) the Conclusions (line 736), and the Methods (lines 982-995, 951-967) the Abstract (lines 45, 57-58) and the Introduction (lines 152-153, 165-166)

(lines 494-563)

Relationship between META, VAR, and measures of integration and segmentation

We have shown that VAR provides superior individual-level classification compared to the conventional metric for metastability, META. However, the classification ability sheds no light onto possible mechanistic explanations as to why this is the case. To address this question, we first need to understand that META and VAR are based on two different order parameters. In dynamical systems theory, an order parameter captures the collective behavior of an underlying high-dimensional non-linear system [39]. META is based on the Kuramoto order parameter [75] which is the mean phase in a system of weakly coupled oscillators. VAR on the other hand, is based on relative phase which has its origins in Superconducting Quantum Interference Device (SQUID) array experiments in the early 1990’s [76]. To compare the relevance of both order parameters, we plot their time-series for one NAP subject, including measures of magnetization ratio, proxy for criticality [62] and chimerality or cluster synchronization [67] in Fig 7.

Fig 7. Time-series for the relative phase order and the Kuramoto order parameter for one NAP subject. A) Time-series for the cosine of relative phase or instantaneous phase-locking for a single subject. B) Time-series for the Kuramoto order parameter or average phase for a single subject. C) Blow-outs showing how relative phase is more informative than average phase for the dynamics of brain activity in one subject. MAG, magnetization ratio; KOP, Kuramoto order parameter, CHI, chimerality.

It can be seen from Fig 7 that taking in-phase and antiphase synchrony into account is more informative on the dynamics of brain activity than just in-phase synchrony, and the magnetization ratio is more relevant to mode switching than chimerality.

Metastable dynamics reflect the competitive tension between global integration and functional segregation [77]. Therefore, any metric or signature of metastability should be correlated with measures of integration and segregation. We calculated the level of global integration and functional segregation as in [78] and compared across groups. In HCPEP a permutation t-test for global integration (GINT) found a statistically significant difference between CON and NAP t(100)=3.70, p<0.001, and for functional segregation (FSEG) t(109)=-3.14, p=0.0023. For Cobre, a statistically significant difference was also found for GINT t(127)=2.96, p=0.0032, and for FSEG t(116)=-3.10, p=0.0026 (see Fig. 8). Additionally, we calculated a Metastability index K, as segmentation/integration, in the spirit of [77]. We performed linear regression of the metrics for global integration, functional segregation, and Metastability Index on META and VAR in HCPEP RUN2 and Cobre. Complete statistical results may be found in S3 Table. For META in HCPEP the highest explanatory power, R2 adjusted, was 0.07 for NAP for global integration and in Cobre 0.55 for CON for global integration. In contrast, for VAR in HCPEP the highest explanatory power, R2 adjusted, was 0.91 for NAP for functional segregation and in Cobre 0.92 for SCHZ for functional segregation. The relationship between META and VAR with these 3 metrics are shown in Fig 8. Based on these results we can infer that VAR is a true measure of the competitive tension between global integration and functional segregation, and therefore a valid signature of metastability.

Fig 8. Relationship between VAR and META with metrics of global integration, functional segregation, and a metastability index K. A) HCPEP global integration versus META. B) HCPEP functional segregation versus META. C) HCPEP Metastability Index versus META. D) Cobre global integration versus META. E) Cobre functional segregation versus META. F) Cobre Metastability Index versus META. G) HCPEP global integration versus VAR. H) HCPEP functional segregation versus VAR. I) HCPEP Metastability Index versus VAR. J) Cobre global integration versus VAR. K) Cobre functional segregation versus VAR. L) Cobre Metastability Index versus VAR. R2 and R2adj results from linear regression. M) HCPEP Group-level differences in global integration. N) HCPEP Group-level differences in functional segregation. O) Cobre Group-level differences in global integration. P) Cobre Group-level differences in functional segregation. GINT, global integration; FSEG, functional segregations; K, Metastability Index.

We have found that metastability as measured with VAR outperforms the conventional measure of META. We have shown that the order parameter of relative phase, as captured by instantaneous phase-locking which reflects in-phase and antiphase synchrony, is more informative than average phase for understanding the dynamics of brain activity. Finally, we have demonstrated that VAR explains between 81-92% of the variance in global integration, functional segregation, and the Metastability Index in the HCPEP and Cobre datasets. In summary, we have shown that antiphase synchrony is not only important for large-scale cortical networks [71], but also for characterizing metastability in healthy controls and groups with a diagnosis of schizophrenia.

(lines 660-677) 

Fourth, VAR explains between 81-92% of the variance in metrics for global integration, functional segregation, and the Metastability Index in both HCPEP RUN2 and Cobre datasets. Although metrics for integration, segregation, and conventional metastability have been estimated previously [78], the explanatory power of metastability on these global metrics of cerebral organization was not investigated. From our study we found that the conventional metric for metastability, META, explained between 0-0.08% of the variance in these metrics in the HCPEP RUN2 dataset, and between 0.16-0.55% in the Cobre data (see S3 Table). For HCPEP this is not surprising as no statistically significant difference was found in global metastability between CON and NAP in RUN2 (see S2 Supporting Information). For CON in Cobre, between 37-55% of the variance in the metrics was explained by metastability, but only 15-18% in SCHZ (see S3 Table). From these results it appears that antiphase synchrony is more prominent in SCHZ.

 Our four complementary perspectives of group-level discrimination, individual-level classification, pathophysiological relevance, and explanatory power, provide preliminary evidence for the face validation of metastability, as measured with VAR, as a candidate neuromechanistic biomarker of schizophrenia. 

(line 736)

classification, pathophysiological relevance, and explanatory power, congruent with

(lines 982-995)

Integration

Global integration was assessed as the connectivity within the time-averaged phase-locking matrix calculated with Eq 1. The matrix is scanned through all possible thresholds from 0 to 1, binarized, and the size of the largest connected component is identified using the Brain Connectivity Toolbox [107]. The integral of the size of the largest connected component as a function of the threshold is taken as an estimate of global integration [78]. 

Segregation

Segregation refers to the decomposition of a system into functional subcomponents and was estimated with the modularity index Q of the time-average phase locking matrix calculated with Eq 1. The Louvain algorithm was used to subdivide the matrix into modules with the Newman modularity Q taken as an estimate of functional segregations [78].

(lines 951 -967)

The Kuramoto order parameter in each community ψ of m regions is given by 

R_ψ (t)=|〈e^(iθ(m,t)) 〉|,_(〖m∈ψ〗_ )

(2)

Metastability was calculated as the standard deviation over time of the Kuramoto order parameter in each community. The mean value of this measure across communities denoted as global metastability, represents the overall variability in phase synchrony across communities.

If we fix time t and estimate the variance of R_ψ (t) across all communities ψ_(1…5), we obtain an instantaneous measure of how chimera-like the system is at time t.

〖CHI〗_ (t)= var(|〈e^(iθ(ψ,t)) 〉| ^ ),_(〖ψ_(1…5)〗_ )

(3)

where CHI is a measure of chimerality, an indicator of cluster synchronization 

(lines 45)

relevance, and explanatory power were assessed using two independent case

(lines 57-58)

Additionally, our new metric showed explanatory power of between 81-92% for measures of integration and segregation.

(lines 152-153)

would tell us about the pathophysiology of schizophrenia; and how well it could explain measures of integration and segregation. 

(lines 165-166)

basal ganglia in early schizophrenia, showed explanatory power of between 81-92% for measures of integration and segregation,

2.5 Is there any difference in metastability due to the choice of resting-state networks? As described in the paper, one can choose pre-defined resting-state network extracted with ICA, network masks or functional template. First, why the authors chose data-driven approach rather than the existing ones? 

We have explained our rational for using a data-driven approach to define communities (lines 301-304)

In contrast to previous studies of metastability in fMRI we choose not to use predefined templates [68], or intrinsic connectivity networks [41] to define our communities. The non-overlapping nature of these networks does not allow flexible allegiance of brain regions to different communities [69].

2.7 As an additional concern, I noted that metastability for each network was computed as the mean value of the variance of instantaneous phase-locking over time in each community/network. Different communities have different number of brain ROIs included in analysis. Synchrony across a large number of ROIs in a community/network will have a smaller variance. In this sense, I am somewhat unsurprised that these networks demonstrate lower levels of both metastability and synchrony in comparison to other networks with smaller number of ROIs. This is a factor that should be included in the statistical analysis to ensure the robustness of this result. This should be discussed in the discussion.

We have included a supporting table S1 Table showing the number of regions assigned to each community for each of the datasets.

We are measuring the variance of instantaneous phase-locking and not in-phase synchrony. The comparison across groups with the largest difference in community brain regions is MODE4 in Cobre with 10 and 37 regions for controls and SCHZ respectively. Despite this difference, VAR in MODE 4 for SCHZ was found to be significantly higher than controls with a moderate effect size as can be seen in S2_Table. Therefore, we do not feel that there is need for concern on this point.

2.9 Second, were there any pre-processing step prior to building a machine learning classifier such as standardization or min-max scaling and so on. 

2.10 Third, when using a naïve Bayes model, how to tune the hyperparameters to obtain optimal parameters that may provide best performance in classification. 

We have updated the text to reflect that no preprocessing took place, and that hyperparameters were not used (lines 438-439).

Other than balancing the samples, we did not perform any preprocessing steps or tune hyperparameters.

2.12 In terms of evaluating the performance of the classifier, the authors used the HCPEP as a training set and the Cobre as a testing set and vice versa for the out-of-sample analysis. Table 4 shows the results of out-of-sample testing for both scenarios. Can you provide more detailed discussion on why the results are different due to the choice of which training sample was used. 

2.14 Further, given a balanced datasets downsampled, the results by classification are not good in terms of accuracy (0.38-0.57) and other measures as well. Why?

We have updated the manuscript to discuss why we believe the choice of training set makes a difference to the classifier performance (line 451-457).

Although VAR in Mode ψ_4 was significantly different between groups in both RUN1 and RUN2, the superior performance of the classifier when trained in RUN2 may be explained by the effect sizes of basal ganglia decouping in that run. Caudate_L (0.435), bilateral Putamen (0.351, 0.357), and Thalamus_L (0.526) showed medium effect sizes only in RUN 2 (see S1 Supporting Information). The poor performance of the classifier when trained in RUN1, RUN3, and RUN4 can be explained by the non-pervasive decoupling of the basal ganglia.

2.13 Overall, a classifier trained on Cobre sample provided better performance than the other case. Any discussion on these results? 

We have updated the manuscript to discuss these results (lines 471-475).

The superior performance of the classifier trained in Cobre can be explained by the different effect sizes of group-level differences in Mode ψ_4 between Cobre and HCPEP. In Cobre the effect size was -0.756 (see S2 Table), whilst the effect sizes in HCPEP varied across runs between 0.037 (RUN3) and 0.399 (RUN2) (see S3 Supporting information).

Reviewer #3

Reviewer #3: In this article, the preprocessed data of HCPEP and Cobre datasets were applied the Hilbert transform, and the authors introduced a new method of Leading Eigenvector Dynamic Analysis (LEiDA) in the frequency domain to capture the specific features of dynamic functional connectivity and then implemented a novel approach to estimate the metastability. Finally, the control group and cases group were classified by a naïve Bayes classifier.

Due to the unclear description of the method, there may be some problems with the results. Some comments are listed as follows:

3.1. There are several places where the data are not clear.

(1) The number of healthy controls and patients in the abstract is not consistent with that in the manuscript. 

We have corrected the demographics Table 7 and noted that subject removal due to incomplete scan or excessive head motion reduced the subject numbers (lines 782-787).

Inspection of the fMRI data for each subject resulted in the exclusion of one subject whose data did not include all 150 volumes. 13 SCHZ subjects with framewise displacement > 0.7mm were also removed. The final dataset therefore used for the Cobre analysis included n=59 SCHZ cases and n=71 HCs. See Table 7 for group demographics and significance of between-group differences in age and sex. 

(2) Line 817, page 36: Which five datasets do they refer to?

We have expanded on the five datasets (line 1048).

In all five datasets (4 HCPEP datasets, 1 Cobre dataset), we assessed the

3.2. Line 702, page 31: “3 conditions x 4 runs” refers to unknown. If “3 conditions” refers to the healthy control group, non-affective psychosis, and schizophrenia respectively, image acquisition in the text only describes that there are 4 runs in the HCPEP dataset, while there are no 4 runs in Cobre dataset for the healthy control group and schizophrenia patients.

We have corrected this error (lines 897-899).

clustered the leading eigenvectors for each of the 10 phase-locked time-series datasets ( HCPEP:CON x 4 runs, HCPEP:NAP x 4 runs, Cobre:CON x 1 run, Cobre:SCHZ x 1 run) with K-means clustering with 300 replications and up to 400

3.3. Line 732 indicates that S7 is included in the supplementary materials, but it is not.

The error in numbering of supporting materials has been corrected.

3.4. The demographic characteristics of participant groups should include the subject information that meets the inclusion criteria of the experiment. 

We have included PANNS, CAINS, and IQ for HCPEP, PANNS and IQ for COBRE in the demographic characteristics (see 3.1 above). Specific DSM classification details were not available for HCPEP.

3.5 In addition, statistical comparisons can be made on information such as gender and age of different groups to show whether there are significant differences between groups in terms of gender and age.

We have included this information in the demographic characteristics (see 3.1 above).

3.6. There are three minor problems.

(1) The full name of fMRI should appear once in the full text, such as the abstract.

(2) The full text does not indicate that iPL is the abbreviation of instantaneous phase-locking.

(3) There are some grammar problems in the article. Please check the grammar carefully. For example, it appears that you are missing a comma after the introductory phrase “In this study”. Consider adding a comma (Line 39, page 3). And it seems that the verb “implements” does not agree with the subject. Consider changing the verb form (Line 49, page 3).

We have addressed these points in the manuscript.

---

## [Decision Letter · Decision Letter 1]

21 Feb 2023

Metastability as a candidate neuromechanistic biomarker of schizophrenia pathology

PONE-D-22-28431R1

Dear Dr. Hancock,

We’re pleased to inform you that your manuscript has been judged scientifically suitable for publication and will be formally accepted for publication once it meets all outstanding technical requirements.

Kind regards,

Daqing Guo

Academic Editor

PLOS ONE

Additional Editor Comments (optional):

Reviewers' comments:

Reviewer's Responses to Questions

**Comments to the Author**

1. If the authors have adequately addressed your comments raised in a previous round of review and you feel that this manuscript is now acceptable for publication, you may indicate that here to bypass the “Comments to the Author” section, enter your conflict of interest statement in the “Confidential to Editor” section, and submit your "Accept" recommendation.

Reviewer #1: All comments have been addressed

Reviewer #2: All comments have been addressed

2. Is the manuscript technically sound, and do the data support the conclusions?

Reviewer #1: Yes

Reviewer #2: Yes

3. Has the statistical analysis been performed appropriately and rigorously? 

Reviewer #1: Yes

Reviewer #2: Yes

4. Have the authors made all data underlying the findings in their manuscript fully available?

Reviewer #1: Yes

Reviewer #2: Yes

5. Is the manuscript presented in an intelligible fashion and written in standard English?

Reviewer #1: Yes

Reviewer #2: Yes

6. Review Comments to the Author

Reviewer #1: (No Response)

Reviewer #2: The revised version of the paper entitled "Metastability as a candidate neuromechanistic biomarker of schizophrenia pathology" was reviewed. Thank you for addressing my comments. I have no further comments and suggestions.

7. PLOS authors have the option to publish the peer review history of their article (what does this mean?). If published, this will include your full peer review and any attached files.

Reviewer #1: No

Reviewer #2: No

---

## [Editor Report · Acceptance letter]

14 Mar 2023

PONE-D-22-28431R1 

Metastability as a candidate neuromechanistic biomarker of schizophrenia pathology 

Dear Dr. Hancock:

I'm pleased to inform you that your manuscript has been deemed suitable for publication in PLOS ONE. Congratulations! Your manuscript is now with our production department. 

Kind regards, 

on behalf of

Dr. Daqing Guo 

Academic Editor

PLOS ONE